# The platelet receptor CLEC-2 blocks neutrophil mediated hepatic recovery in acetaminophen induced acute liver failure

Abhishek Chauhan [1✉], Lozan Sheriff [1], Mohammed T. Hussain [1], Gwilym J. Webb[1], Daniel A. Patten [1], Emma L. Shepherd[1], Robert Shaw[2], Christopher J. Weston [1], Debashis Haldar [1], Samuel Bourke[1], Rajan Bhandari[1], Stephanie Watson[3], David H. Adams[1], Steve P. Watson [3,4] & Patricia F. Lalor [1]

Acetaminophen (APAP) is the main cause of acute liver failure in the West. Specific efficacious therapies for acute liver failure (ALF) are limited and time-dependent. The mechanisms that drive irreversible acute liver failure remain poorly characterized. Here we report that the recently discovered platelet receptor CLEC-2 (C-type lectin-like receptor) perpetuates and worsens liver damage after toxic liver injury. Our data demonstrate that blocking platelet CLEC-2 signalling enhances liver recovery from acute toxic liver injuries (APAP and carbon tetrachloride) by increasing tumour necrosis factor-α (TNF-α) production which then enhances reparative hepatic neutrophil recruitment. We provide data from humans and mice demonstrating that platelet CLEC-2 influences the hepatic sterile inflammatory response and that this can be manipulated for therapeutic benefit in acute liver injury. Since CLEC-2 mediated platelet activation is independent of major haemostatic pathways, blocking this pathway represents a coagulopathy-sparing, specific and novel therapy in acute liver failure.

[1] Centre for Liver and Gastrointestinal Research, Institute of Immunology and Inflammation, and National Institute for Health Research (NIHR) Birmingham Biomedical Research Centre, The Medical School, University of Birmingham, Birmingham B15 2TT, UK. [2] Technology Hub, College of Medical and Dental Sciences, University of Birmingham, Birmingham B15 2TT, UK. [3] Institute of Cardiovascular Sciences, The Medical School, University of Birmingham, Birmingham B15 2TT, UK. [4] Centre of Membrane Proteins and Receptors (COMPARE), Universities of Birmingham and Nottingham, The Midlands, Nottingham, UK. ✉email: a.chauhan.1@bham.ac.uk

Toxic liver injury exemplified by acetaminophen (APAP) overdose is the leading cause of acute liver failure (ALF) in Europe and North America[1]; it is responsible for more cases of ALF than all other aetiologies combined[2]. Acute liver failure has a high mortality rate[3] and accounts for 10% of all liver transplants in the UK. Although N-acetyl cysteine (NAC) can prevent liver injury if given in time, once liver failure is established the only treatment other than transplantation is supportive[4,5]. Thus, there is an unmet clinical need for effective and specific treatments to reduce or reverse acute toxic liver injury.

Hepatic injury in APAP overdose-induced acute liver failure in humans follows a characteristic pattern with biochemical signs of liver damage becoming apparent at 24–48 h[6], and hepatocyte death peaking at around 72 h[7]. In severe injury, this results in fulminant liver failure necessitating superurgent liver transplantation. Murine APAP toxicity closely mimics the human response and is a valuable model for preclinical studies investigating toxic liver damage[8]. APAP in both humans and mice is metabolized by cytochrome P450 isoforms into the alkylating metabolite N-acetyl-p-benzoquinone imine (NAPQI)[6]. Excess NAPQI formation in APAP overdose, overwhelms the livers antioxidant glutathione (GSH) reserves. This allows excess NAPQI to bind mitochondrial proteins, forming cytotoxic adducts[9] that drive severe mitochondrial dysfunction and extensive hepatocellular necrosis[10]. Another agent extensively studied in the context of acute toxic murine liver injury is carbon tetrachloride (CCl$_4$), which is also metabolized by cytochrome P450 enzymes into reactive intermediates[11,12]. These radicals cause hepatocyte membrane lipid peroxidation and extensive hepatocellular necrosis leading to liver failure.

Regardless of the aetiology[13], hepatocyte necrosis in response to toxic injury results in the release of damage associated molecular patterns (DAMPs) from dying hepatocytes[14]. These activate the innate immune system in what is termed 'sterile inflammation' (SI)[14]. SI is critical to liver healing and recovery but if excessive can exacerbate liver injury and drive fulminant liver failure[13]. Hepatic neutrophil infiltration represents a key part of the innate immune response to acute liver injury[14,15] and is essential for liver recovery after a toxic liver insult[16]. The signals that dictate whether the outcome of injury is regeneration and healing on one hand, or progressive inflammatory damage on the other are poorly understood. Elucidation of these mechanisms could lead to novel treatments that reduce injury and enhance hepatic recovery after a toxic injury.

An increasing body of recent literature recognizes that platelets play a central but paradoxical role in liver injury, being able to both drive damage and aid resolution[17]. Platelets interact with the key protagonists of the sterile inflammatory response including neutrophils and macrophages[14] and enhance hepatic leucocyte recruitment into the liver across the sinusoidal endothelium[18]. Kupffer cells (liver resident macrophages) recognize. DAMP expression and upregulate cytokines central to the initiation of SI[14]. Neutrophils infiltrate the liver early after SI and are implicated in mediating resolution from acute toxic liver injury after CCl$_4$ induced[19] and thermal liver injury[16]. Whilst platelets have been shown to accumulate within the liver during APAP mediated liver toxicity and platelet depletion attenuates liver injury[20]; little is actually known about the molecular basis of platelet function during acute liver failure and specifically whether manipulating platelet function influences the sterile inflammatory response to toxic liver injury.

Platelet activation occurs through multiple pathways including ligation of platelet-expressed C-type lectin-like receptor (CLEC)-2 by its ligand podoplanin[21]. As inflammatory macrophages upregulate podoplanin[22,23] and have an important role

in driving the sterile inflammatory response to toxic liver injury[14], we investigate whether CLEC-2 signalling driven by inflammatory macrophages, including Kupffer cells and infiltrating macrophages, plays a role in acute toxic liver injury. Importantly as this platelet signalling pathway is independent of major haemostatic pathways[24] it potentially represents an attractive avenue for the development of new therapies in liver injuries that are often associated with coagulation impairments. We present data in mice and humans demonstrating that abrogation of CLEC-2 mediated platelet signalling enhances recovery from acute liver injury through TNF-α dependent hepatic recruitment of neutrophils.

## Results

**Podoplanin is upregulated during liver injury and platelets sequester to podoplanin expressing macrophages.** Podoplanin, the cognate ligand for the platelet activating receptor CLEC-2 is minimally expressed in the uninjured liver in both mice and humans with expression primarily confined to periportal lymphatic vessels (Fig. 1a, b). In contrast, human and murine livers respond to acute liver injury by increasing hepatic podoplanin expression (Fig. 1a-APAP, 1b-CCl$_4$ [mice] and APAP, c–f [human]). Immunofluorescent podoplanin detection demonstrated that during human acute APAP driven liver injury, enhanced podoplanin expression was seen upon hepatic macrophages within pericentral areas (Fig. 1c, Supplementary Fig. 1) (where maximum APAP induced injury is seen) and also in periportal areas (Fig. 1c ii). To establish the nature of the podoplanin expressing macrophages we next phenotyped macrophages from acutely injured murine livers[25] and found that these were a combination of infiltrating and resident (Kupffer) cells (Fig. 1d). The increase in podoplanin mRNA expression within APAP injured livers compared to non-injured control human liver tissue was confirmed by PCR (Fig. 1e). To investigate whether platelets are exposed to the increase in podoplanin in acute liver injury, we stained for the platelet specific integrin subunit CD41. Figure 1f shows that CD41 positive platelets colocalize with podoplanin-positive hepatic macrophages during acute liver injury in mice (Fig. 1f, upper row) and humans (Fig. 1f, lower row).

**Blocking platelet CLEC-2 signalling accelerates liver healing after toxic liver injury in mice.** After establishing that hepatic expression of podoplanin, the only known endogenous activating ligand for CLEC-2, is increased after acute liver injury and that platelets sequester to podoplanin expressing macrophages, we investigated the functional consequences of blocking CLEC-2-podoplanin interaction during acute liver damage. For this we used Pf4CreCLEC1b$^{fl/fl}$ mice, that have deletion of CLEC-2 gene (CLEC1b) in cells of the megakaryocytic lineage[26]. Hepatic necroinflammation due to APAP or CCl$_4$ is dependent on conversion to hepatotoxic metabolites by cytochrome P450 CYP2E1 which is unaffected in the Pf4CreCLEC1b$^{fl/fl}$ mouse (Supplementary Fig. 2). We observed liver injury indicated by increased transaminase levels in serum as early as 6 h after treatment in APAP treated mice and by 24 h in CCl$_4$ treated mice (Fig. 2a, b). There was no difference in the amount of hepatic damage observed when comparing WT (wild type) and CLEC-2-deficient mice at these early time points (Fig. 2a, b). However, at subsequent time points (APAP—24 and 48 h; CCl$_4$—48 and 72 h), CLEC-2-deficient mice exhibited significantly lower serum ALT activity compared to WT control mice which developed worsening liver damage over time. This was supported by histological staining at the time of maximal damage in WT mice (48 h—CCl$_4$, or 24 h—APAP) as shown in Fig. 2c, Control mice that had been given PBS or mineral oil alone exhibited no liver injury (Fig. 2a, b, d). We

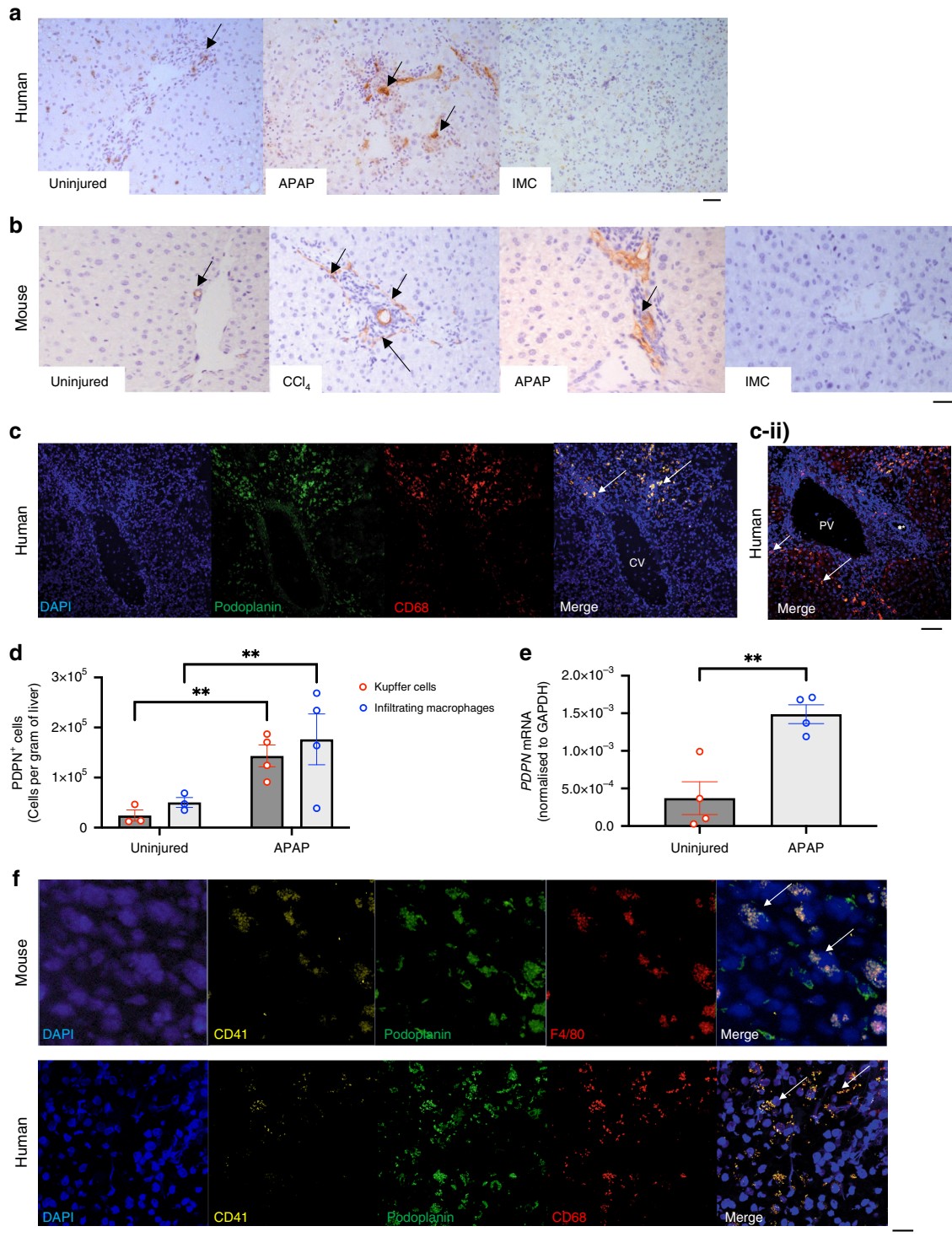

also noted that a reduced number of platelets were sequestered to macrophages in peri-central areas in CLEC-2 deficient mice compared to WT mice (Fig. 2d). Consistent with this, platelet P-selectin staining was reduced in CLEC-2-deficient mice (Fig. 2e, Supplementary Fig. 3).

**Acute liver injury is exacerbated by podoplanin-mediated CLEC-2 signalling**. To investigate whether podoplanin-mediated CLEC-2-dependent platelet activation perpetuated liver injury, we used a transgenic podoplanin-deficient mouse model or a podoplanin-blocking antibody. Mice with conditional deletion for the gene encoding podoplanin expression on haematopoietic cells (Vav1-iCre$^+$PDPN$^{fl/fl}$) and WT animals had similar levels of liver injury at early time points, but in agreement with data from CLEC-2-deficient mice, at the time of maximal liver damage (Fig. 2a, b) podoplanin-deficient mice exhibited less hepatic damage as shown by measurement of serum ALT activity (Fig. 3a) Similarly, treatment of WT mice with a function-blocking antibody directed

**Fig. 1 Podoplanin is upregulated within the liver during liver injury and platelets sequester to podoplanin expressing macrophages. a** Podoplanin expression in brown (black arrows) in acetaminophen (APAP) injured and non-injured human liver (IMC-isotype matched control antibody) ×10 magnification (scale bar—100 µm). **b** Podoplanin expression in brown (black arrows) in injured (APAP 24 h; CCl4 48 h) and uninjured mouse livers is shown (arrows) ×20 magnification, IMC-isotype matched antibody staining (scale bar—200 µm). **c** Confocal microscopy pictures of sections from an acutely injured APAP human liver (DAPI nuclear stain-blue, podoplanin-green, macrophages-CD68-red) near a pericentral area (CV-central vein) ×10 magnification (scale bar—100 µm). **c** ii Confocal microscopy pictures of sections from an acutely injured APAP human liver (DAPI nuclear stain-blue, podoplanin-green, macrophages-CD68-red); top row shows podoplanin expressing macrophages (white arrows) near a portal tract (PV-portal vein, asterisk-bile duct). **d** Wild type mice were injected with IP APAP (350 mg/kg)] or PBS, livers were removed at 24 h, digested and podoplanin expressing infiltrating macrophages (defined by the gating strategy CD45$^+$ CD3$^-$CD11b$^{hi}$F4/80$^+$) and resident Kupffer cells (defined by the gating strategy CD45$^+$ CD3$^-$CD11b$^{int}$F4/80$^+$) were isolated (gating strategy Supplementary Fig. 1), quantified by flow cytometry and expressed as number of cells per gram of liver tissue ($n = 4$, in injury group), from mice. **e** qPCR data indicating podoplanin is upregulated within the liver after human acute drug induced liver injury (DILI) compared to normal liver control (NL)- *PDPN* mRNA (normalized to *GADPH*) differences assessed using either unpaired Student's *t* test or a Mann–Whitney test *$P < 0.05$, **$P < 0.01$, ***$P < 0.001$. Each data point is representative of an independent animal and data is presented as mean ± SEM. **f** High magnification (×40 or ×63) confocal images from APAP injured mouse and human livers demonstrating platelet (CD41-yellow) sequestration to podoplanin (green) expressing macrophages [F4/80 (mouse), CD68 (human)—both red] (white arrows). (scale bar—200 µm). Confocal images captured using a Zeiss LSM880 Airyscan detector, set to SR mode and with Zeiss LSM image software; brightfield images acquired at ×10, ×20 or ×40 magnification using a Leica CTR6000 microscope with QCapture software. All images are representative of at least six livers.

against podoplanin, reduced liver injury at time points of maximal liver injury (Fig. 3a) compared to isotype-matched antibody-treated controls. In support of this, histological analysis of sections from these mice confirmed that WT or mice treated with the isotype matched control antibody had greater amounts of pericentral necrosis after CCl4 or APAP administration compared to mice that had a genetic or antibody mediated deficiency in podoplanin (Fig. 3b).

**Blocking the CLEC-2 podoplanin axis enhances neutrophil recruitment to the injured liver.** To explain how inhibiting platelet CLEC-2 signalling enhanced recovery from CCl4 or APAP driven injury, we investigated the nature of the sterile inflammatory response. Neutrophils are known to mediate liver healing after sterile liver injury by clearing cellular debris through phagocytosis[16]. Using flow cytometric assessment of liver infiltrating immune cell populations we observed a significantly greater number of neutrophils (Fig. 4a, b) infiltrating the injured livers of CLEC-2- or podoplanin-deficient, or podoplanin antibody-treated mice after administration of APAP or CCl4 compared to the livers of the corresponding control mice. This was confirmed by immunostaining, Fig. 4c thus shows enhanced numbers of elastase-positive neutrophils after liver injury within the livers of CLEC-2/podoplanin deficient mice. Of note, the total numbers of infiltrating leucocytes (see gating strategy for CD45$^+$ cells, myeloid cells and lymphocytes shown in Supplementary Figs. 4 and 5) were similar between WT and CLEC-2-deficient mice at each time point, regardless of type of injury (Supplementary Fig. 6). We next measured neutrophil function following acute liver injury and found that neutrophils from CLEC-2-deficient mice exhibited increased phagocytic ability compared to those from WT mice (Fig. 4d and Supplementary Fig. 1). This suggested that interfering with the CLEC-2/podoplanin axis not only enhanced neutrophil recruitment to the injured liver but also the ability of those neutrophils to phagocytose cellular debris. To confirm that the reparative response was indeed neutrophil driven, we administered a selective neutrophil depleting anti-Ly6G antibody to CLEC-2-deficient mice prior to CCl4 or APAP administration. As expected, the antibody reduced hepatic neutrophil infiltration during acute liver injury (Fig. 4e) by depleting circulating neutrophils (Fig. 4f); this then abrogated the protective effect of CLEC-2 deficiency (Fig. 4g) to toxic liver injury. Increased hepatic regeneration did not contribute to this observed phenotype (Supplementary Fig. 7).

**Neutrophil recruitment in CLEC-2-deficient animals is dependent on macrophage TNF-α production.** We have described increased hepatic expression of podoplanin and platelet sequestration to podoplanin expressing macrophages during acute liver injury. To investigate the functional consequences of this, we cocultured WT macrophages isolated from murine liver in LPS (Supplementary Fig. 8) with platelets. Incubation of podoplanin-expressing macrophages with CLEC-2 deficient platelets was associated with increased TNF-α production relative to WT platelets (Fig. 5a). To extend these findings in vivo, we stained for TNF-α in hepatic macrophages and our cytometric assessment showed that APAP injury induced TNF-α expression within hepatic macrophages (Fig. 5b). Although the amount of TNF-α expression per macrophage (Fig. 4b) was similar between macrophages from WT and CLEC-2 deficient mice after APAP injury (Fig. 5b), we observed that there was a significantly increase in the total number of TNF-α expressing, podoplanin-positive infiltrating macrophages in CLEC-2 deficient mice (Fig. 5c). These data suggest that the increase in the number of podoplanin-expressing macrophages in CLEC-2 deficient mice drives the increase in total TNF-α production after an acute liver injury. Consistent with this, CLEC-2 deficient mice had significantly higher serum TNF-α levels compared to WT mice after toxic liver injury (Fig. 5d). To investigate whether the neutrophil recruitment we observed in the CLEC-2 deficient mice was TNF-α dependent we used etanercept to inhibit TNF-α prior to induction of liver injury. This restored liver injury to WT levels (Fig. 5e) by inhibiting neutrophil recruitment to the injured liver in CLEC-2-deficient mice (Fig. 5f).

These data support a model in which CLEC-2-or podoplanin deficiency mediates its hepatoprotective effect through enhanced macrophage driven TNF-α production and recruitment of neutrophils.

## Discussion

Platelets have been demonstrated to play important roles in pathophysiological processes relevant to liver disease[27,28]; however, as these can be either detrimental or protective dependent on the context[17] their overall contribution to liver pathology is unclear and is likely to be disease and stage-dependent[28]. We report a mechanism through which platelet signalling through CLEC-2 amplifies acute liver injury in response to APAP and CCl4 toxicity. We demonstrate that selectively abrogating platelet CLEC-2 signalling accelerates early recovery from toxic liver injury by increasing hepatic neutrophil function and infiltration

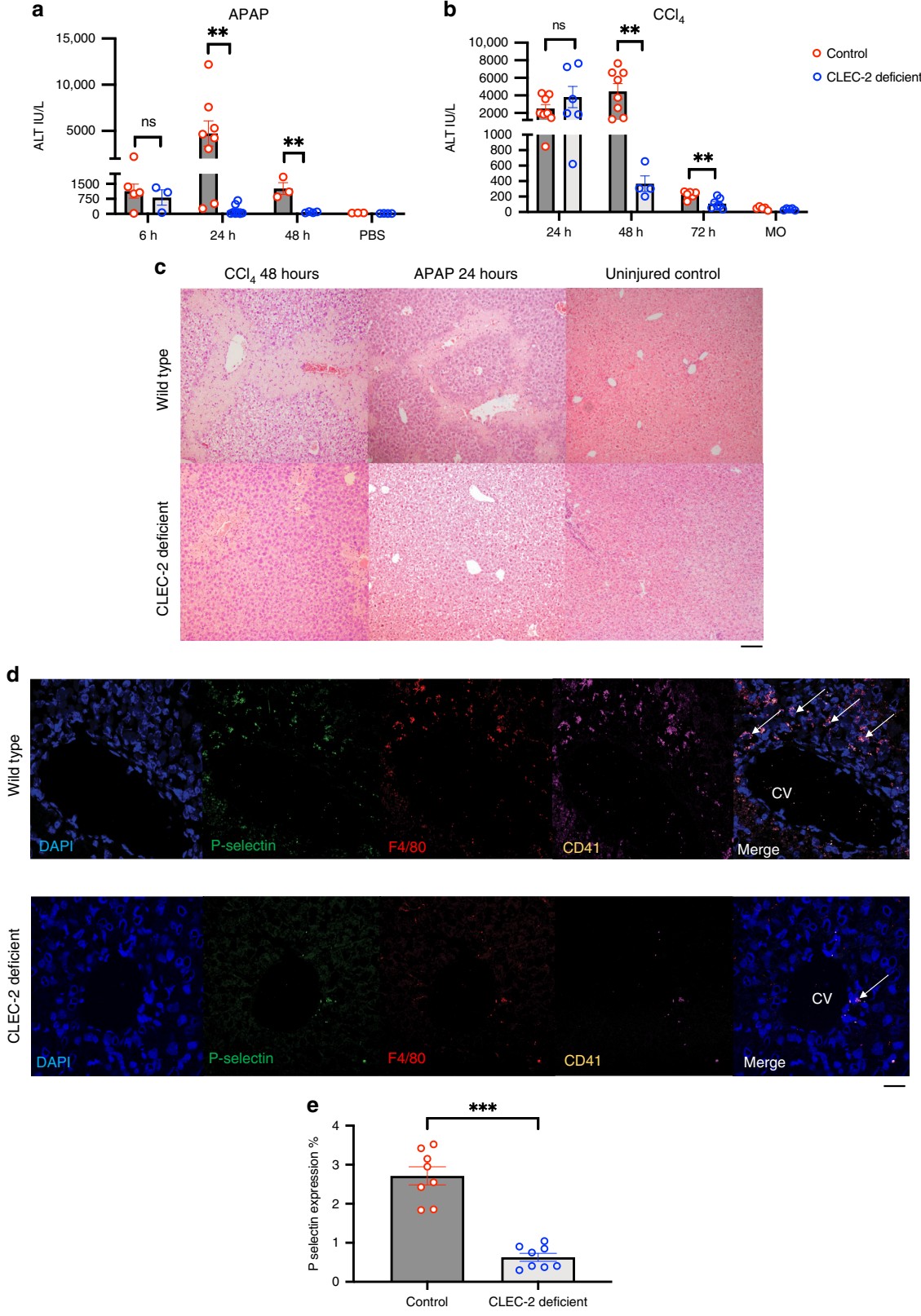

through a TNF-α driven mechanism. Importantly as CLEC-2 mediated platelet activation is independent of major haemostatic pathways[21], the bleeding risk of CLEC-2 blockade is minimal[23]; this remains an important consideration given the bleeding diathesis often present in acute liver failure[29–32].

We found that acute toxic injury to the liver results in increased podoplanin expression on both resident and infiltrating liver macrophages. Under resting or non-inflammatory conditions there is minimal podoplanin expression in the liver. The increased availability of podoplanin in the acutely injured liver could provide the ligand to activate the CLEC-2 pathway within platelets, which were noted to sequester to podoplanin-expressing macrophages in acute liver injury in both mice and humans. Upregulation of podoplanin on macrophage populations has been

**Fig. 2 Mice with CLEC-2 deficient platelets (*CLEC1b*^fl/fl^Pf4Cre) exhibit reduced hepatic platelet sequestration and enhanced liver healing after a toxic injury.** Wild type (Cre negative) or CLEC-2 deficient (*CLEC1b*^fl/fl^Pf4Cre) Mice were injected with IP CCl₄(1 mg/kg) or IP APAP (350 mg/kg) and sacrificed 24, 48 or 72 h (IP CCl4) or 6, 24, 48 h (IP APAP) after injection. For uninjured controls, (both wild type and CLEC-2 deficient) mice were injected with mineral oil (MO) or phosphate buffered saline (PBS) alone. **a** Serum alanine transaminase activity at different time points comparing wild type with *CLEC1b*^fl/fl^Pf4Cre mice injected with IP APAP ($n = 4$–8 at point of peak injury). **b** Serum alanine transaminase activity at the different time points comparing wild type with *CLEC1b*^fl/fl^Pf4Cre mice injected with IP CCl₄ ($n = 4$–8). Representative non-injured controls (PBS-APAP, MO-CCl₄) from peak point of injury in each model (APAP-24 h and CCl4-48 h) are shown. **c** Representative images from formalin fixed liver sections stained with hematoxylin and eosin (48 h after either CCl4 or APAP injection or non-injured controls). Confocal images captured using a Zeiss LSM880 Airyscan detector, set to SR mode and with Zeiss LSM image software; brightfield images acquired at ×10, ×20 or ×40 magnification using a Leica CTR6000 microscope (Leica, UK) with QCapture software, all images representative of at least six livers (scale bar—100 μm). **d** High magnification (×40) confocal microscopy images of sections from acutely injured mouse livers demonstrating both reduced platelet sequestration in CLEC-2 deficient mice (bottom row) when compared to wild type mice (top row) in pericentral areas (CV-central vein) (DAPI nuclear stain-blue, P-selectin-green, macrophages-F4/80-red, platelets-CD41-purple) (scale bar—200 μm). **e** Percentage area of P selectin expression measured in murine liver sections (Supplementary Fig. 3) comparing wild type mice with CLEC-2 deficient mice after toxic liver injury. Percentage area measured on eight randomly selected fields from either CLEC-2 deficient or WT mice, measured using image J version 1.8 (National Institutes of Health and the Laboratory for Optical and Computational Instrumentation, University of Wisconsin). Differences assessed using either unpaired Students $t$ test or a Mann–Whitney test *$P < 0.05$, **$P < 0.01$, ***$P < 0.001$. Each data point is representative of an independent animal and data is presented as mean ± SEM.

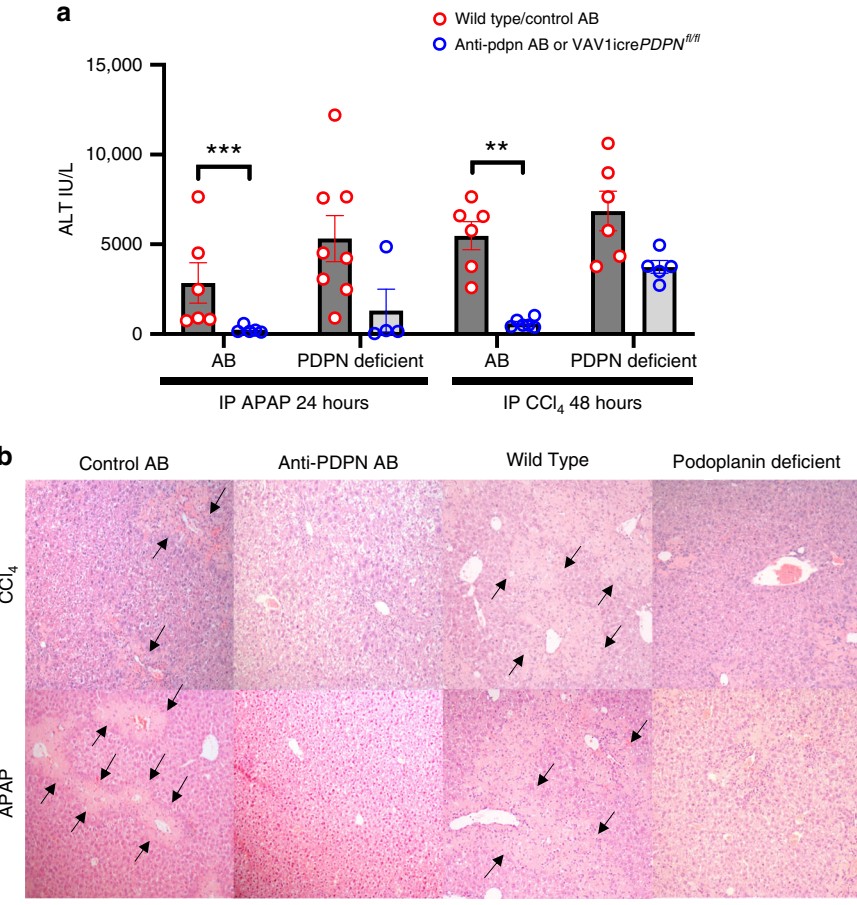

**Fig. 3 Removing podoplanin removes the signal for platelet CLEC-2 signalling and enhances liver healing after CCl4 or APAP induced liver injury.** WT (Cre negative) or podoplanin deficient (Vav1-iCre+ pdpn^fl/fl^) mice were injected with IP CCl₄ (1 mg/kg) and sacrificed at 48 h or IP APAP (350 mg/kg) and sacrificed at 24 h. **a** Serum ALT level is shown from mice treated with either APAP (acetaminophen) or CCl₄, the first and third pairs of columns compare WT mice that were pretreated with either a podoplanin function blocking antibody (white columns) or the corresponding isotype matched control antibody (black columns) ($n = 6$–8). The second and fourth column pairs compare genetically deficient podoplanin mice (Vav1-iCre+PDPN^fl/fl^-white columns) and WT controls-black columns ($n = 4$–8). **b** Representative hematoxylin and eosin stained sections from the time points mentioned above are shown, images acquired at ×10 magnification using a Leica CTR6000 microscope with QCapture software, images representative of at least four different mice in each group (scale bar—100 μm). Black arrows highlight areas of hepatic necrosis. Differences assessed using either unpaired Students $t$-test or a Mann–Whitney test *$P < 0.05$, **$P < 0.01$, ***$P < 0.001$. Each data point is representative of an independent animal and data is presented as mean ± SEM.

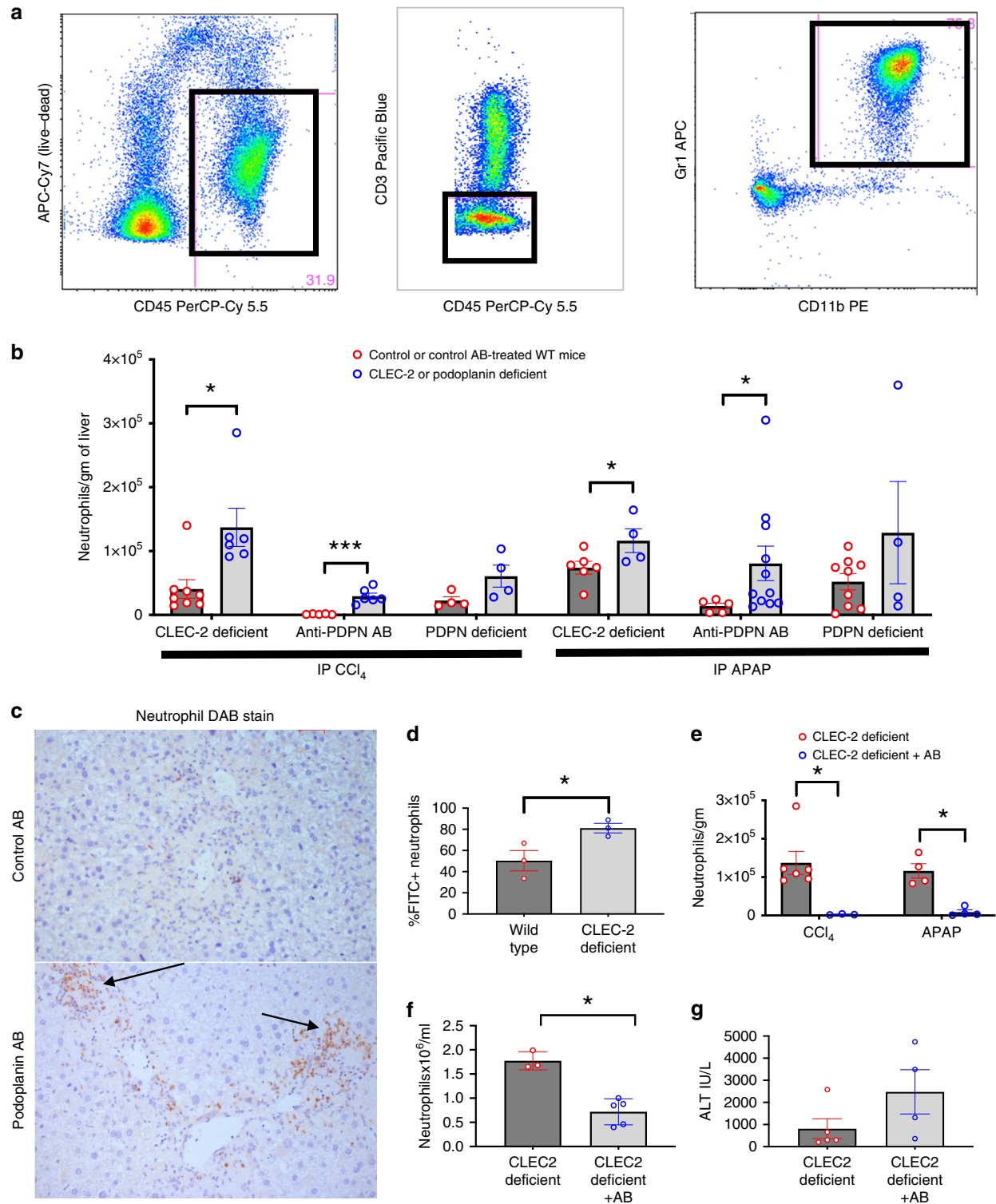

described in the context of peritonitis[33], and in the liver during sepsis driven by IFNγ and TLR4[21]. This fits with the increased gut permeability, bacterial translocation[34] and the consumptive coagulopathies that characterize acute liver injury[1]. Multiple studies have established the role of CLEC-2 in driving platelet activation[26,35]; investigators have previously demonstrated both how inflammatory macrophages upregulate and activate platelets via CLEC-2 in vitro[22] and also the importance of this pathway in modulating the outcome from hepatic inflammation[21,23]. We use P-selectin to confirm intrahepatic platelet activation after an acute

liver injury and whilst we note lower amounts of P-selectin expression in mice that are CLEC-2 deficient it is noteworthy that P-selectin is stored within platelet granules and a degree of constitutive expression is therefore unavoidable. This limitation exists with other platelet granule stored markers of platelet activation including TREM-like transcript 1(TLT-1)[36].

Enhanced liver recovery was associated with significantly increased circulating levels of TNF-α in the CLEC-2-deficient mice. Hepatic macrophage production of TNF-α has been shown to mediate hepatic regeneration in hepatectomized mice[37-39],

**Fig. 4 Blocking CLEC-2-mediated platelet activation enhances hepatic neutrophil accumulation.** Livers from mice sacrificed at the above time point of peak injury [48 h after CCl$_4$ (1 mg/kg) and 24 h after APAP (350 mg/kg)] were digested and neutrophils were isolated, quantified by flow cytometry and expressed as number of cells per gram of liver tissue from mice with either a genetic CLEC-2 (*CLEC1b*$^{fl/fl}$Pf4Cre) or podoplanin (Vav1-iCre$^+$ *pdpn*$^{fl/fl}$) deficiency or an antibody mediated podoplanin blockade and the corresponding controls. **a** Our strategy to define live gated CD45$^+$Cd3$^-$CD11b$^+$Gr1$^+$ cells (neutrophils) is shown. **b** number of neutrophils per gram of liver tissue at the time points mentioned above comparing either control WT and CLEC-2-deficient mice or control WT and podoplanin-deficient or mice that had been treated with a function blocking podoplanin antibody and corresponding isotype matched controls are shown ($n = 4-9$). **c** Liver sections were stained with neutrophil elastase DAB stain (arrows), representative fields from anti-podoplanin antibody (bottom picture) and isotype antibody (top figure) treated mice are shown at ×20 magnification acquired using a Leica CTR6000 microscope with QCapture software, images representative of six different mice in each group; scale bar = 100 μm. **d** Neutrophils from whole mouse blood harvested at peak time point of liver injury after APAP administration were assessed for their ability to phagocytose FITC labelled bacteria (Supplementary Fig. 1) as per manufacturer's instructions. Data shows % of neutrophils that had ingested FITC labelled bacteria and compares neutrophils from WT mice with CLEC-2 deficient mice. CLEC-2 deficient mice were pre-treated with a specific neutrophil depleting antibody prior to either CCl$_4$ or APAP injection, $n = 3$. **e** hepatic neutrophil infiltration in CLEC-2 deficient mice after APAP or CCl$_4$ injection is shown ($n = 4-6$). **f** Peripheral circulating neutrophil count ($n = 3-5$) and **g** serum ALT activity in CLEC-2 deficient comparing mice that had been treated with either the neutrophil depleting antibody or sham after acute liver injury is shown ($n = 4-5$). Differences assessed using either an unpaired students $t$ test or a Mann–Whitney test *$P < 0.05$, **$P < 0.01$, ***$P < 0.001$. Each data point is representative of an independent animal and data is presented as mean ± SEM.

abrogating TNF-α function in these models inhibits liver recovery[40] and thus reduces survival. In human acute liver failure, monocyte derived TNF-α has a clearly described protective role, with low TNF-α levels due to monocyte exhaustion resulting in poor outcomes[41]. Our in vitro experiments showed heightened TNF-α production in CLEC-2 deficient platelet-macrophage assays compared to WT platelet-macrophage assays. Lectin receptors are important[42] in allowing hepatic macrophages to clear platelets from the circulation[42,43]; a CLEC-2 deficiency may alter this process thereby minimizing exhaustion and maintaining ability to secrete cytokine; this is consistent with our findings. The underlying mechanism through which platelet CLEC-2 deficiency alters cytokine production warrants further investigation. In vivo we noted an increase in the number of podoplanin-positive macrophages expressing TNF-α in CLEC-2 deficient mice compared to WT mice after acute liver injury. This could be due to a failure of infiltrating macrophages to egress from the inflamed liver in a timely fashion. Indeed there is a previously described role for the CLEC-2/podoplanin axis in facilitating the egress of other antigen presenting cells from lymphoid organs[44] and stromal cells from areas of inflammation[45]. Taken together these data suggest that activation of CLEC-2 signalling within platelets by macrophage expressed podoplanin within the acutely injured liver suppresses TNF-α release by those macrophages. Blocking this activation accounts for the increase in circulating TNF-α levels seen in CLEC-2 deficient mice after acute liver injury. Although TNF-α is produced by multiple cell types during acute liver injury, our in vitro and in vivo experiments suggest that at least part of this is derived from hepatic macrophages.

Platelet interactions with macrophages are known to modulate macrophage function, differentiation and cytokine secretion[46–48]; the molecular pathways that mediate these effects are poorly understood. This is the first report of a molecular pathway (CLEC-2/podoplanin) that links platelet function with an alteration of macrophage cytokine responses, enhancement of neutrophil recruitment and function and ultimately how this mediates restorative effects in acute liver injury. It is prudent to however note that while TNF-α is a pleiotropic cytokine having varied roles in liver injury, exhibiting hepatoxicity in certain situations and aiding recovery in others[49], it appears that in acute toxic liver injuries the latter role is more significant; this may explain in part the adverse outcomes seen in patients with severe alcoholic hepatitis treated with anti-TNF-α therapy[50,51].

Blocking platelet CLEC-2 signalling by inducing a selective platelet CLEC-2 deficiency or a podoplanin blockade resulted in a reduction in hepatocellular damage and transaminase release through increased hepatic neutrophil recruitment. Whilst

authors[52,53] have previously argued that neutrophils actively drive APAP induced liver injury, these studies rely upon neutrophil depleting antibodies which have marked off target effects[54]. Direct neutrophil driven hepatotoxicity was demonstrated using hepatoma cells in vitro which have unclear relevance to the in vivo sterile injury that APAP elicits[52,54]. Importantly whilst numerous studies have shown that APAP induced liver damage is not driven by neutrophils[55,56], it is clear that neutrophil activation and hepatic infiltration occurs after APAP induced liver damage[16,57,58]. This neutrophil infiltration is increasingly thought to represent the innate immune reparative response designed to aid a return to hepatic homoeostasis after acute injury[15,59,60]. Our data demonstrate that platelet CLEC-2 deficiency enhances both hepatic neutrophil infiltration and neutrophil phagocytic function. Based on our ability to abrogate hepatic neutrophil recruitment with etanercept, it is likely that heightened TNF-α production plays a critical role in hepatic recruitment of neutrophils in CLEC-2-deficient mice. This supports previous reports of the neutrophil 'function' enhancing role of TNF-α[61], including phagocytosis, which then plays a fundamental role in halting liver damage and driving hepatic recovery in acute liver injury[16,62]. Neutrophils thus mediate liver healing after APAP induced liver injury[51] with defective neutrophil function resulting in poor outcomes[63–65]. This has direct relevance to the clinic where enhancing neutrophil numbers and function using granulocyte-colony stimulating factor (G-CSF) improves survival in patients with acute liver failure[66] and severe alcoholic hepatitis[67]. Although our data suggest that neutrophil recruitment is driven by hepatic macrophage produced TNF-α, additional platelet–neutrophil–endothelium interactions may also have a role. Specifically, platelet attachment to the sinusoidal endothelium[18], formation of circulating neutrophil: platelet aggregates or enhanced NET formation[68] could also contribute to our observations of neutrophil mediated liver recovery in CLEC-2 deficiency. Finally, it is also possible that enhanced TNF-α production we observed with CLEC-2 deficiency resulted in increased recruitment or activation of Cd11b$^+$ myeloid-derived suppressor cells[69] that could then accentuate the negative feedback regulation of hepatic inflammation after exposure to hepatotoxicant[70,71].

In conclusion, during acute liver injury hepatic macrophages upregulate podoplanin; the resultant CLEC-2 driven platelet activation dampens TNF-α production which then impedes restorative neutrophil mediated liver repair. By interfering with the CLEC-2/podoplanin axis, we provide the first demonstration of a mechanism to enhance neutrophil mediated reparative sterile inflammation to drive efficient and rapid liver recovery. Given the

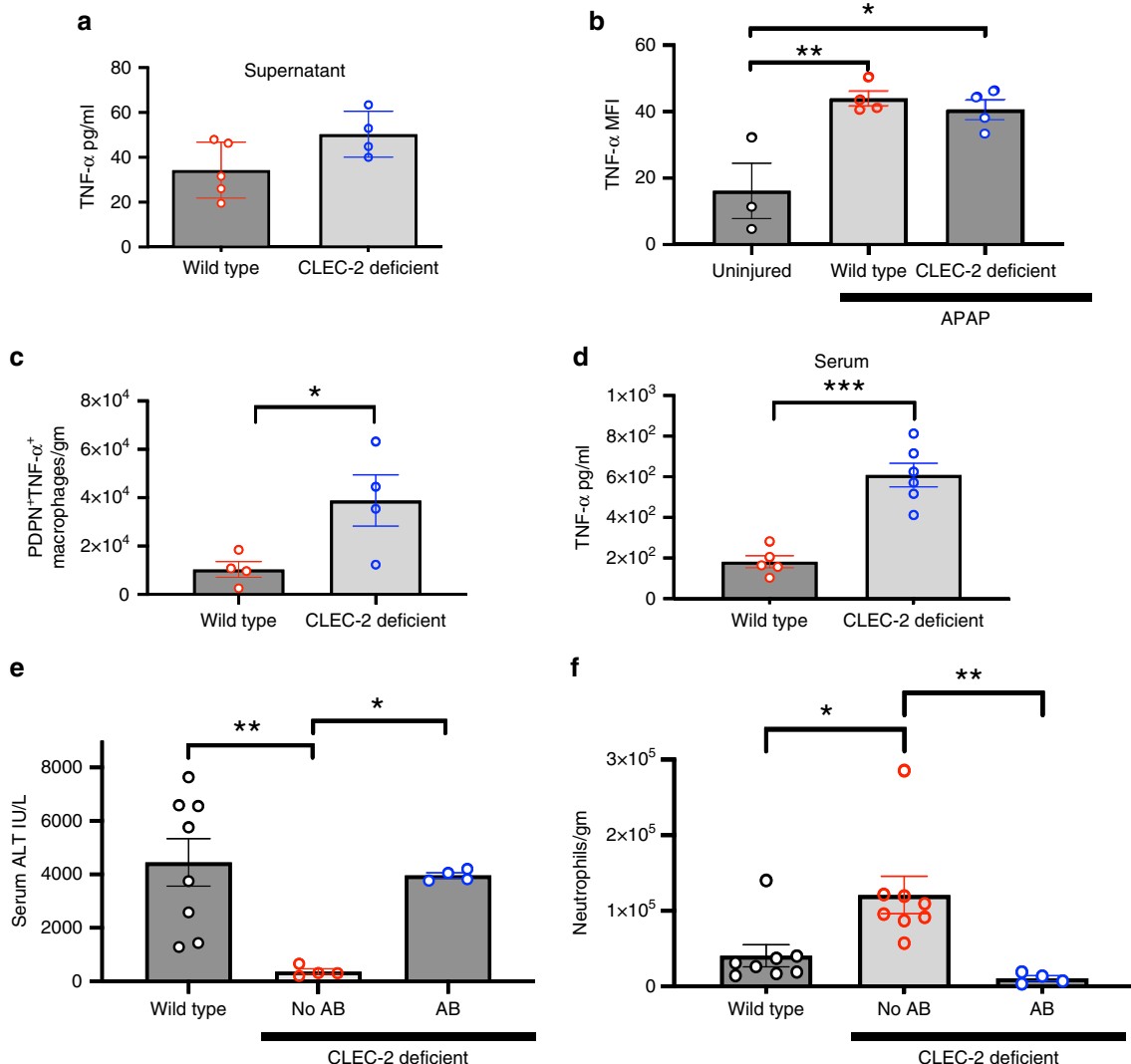

**Fig. 5 Macrophage produced TNF-α recruits neutrophils after toxic injury. a** TNF-α concentration in supernatant from in vitro LPS stimulated wild type murine macrophages cultured with either WT or CLEC-2 deficient platelets (data from six individual mice in each group. **b** Wild type or CLEC-2 deficient (PF4CreCLEC1b^fl/fl mice) were injected with IP APAP (350 mg/kg)] or PBS. Livers were removed at 24 h, digested and then macrophages isolated (defined by the gating strategy CD45+CD3−CD11b+F4/80+) and then using flow cytometric analysis TNF-(mean fluorescent intensity (MFI) was calculated per macrophage (n = 4 in APAP treated mice). **c** Wild type or CLEC-2 deficient (PF4CreCLEC1b^fl/fl mice) were injected with IP APAP (350 mg/kg)]. Livers were removed at 24 h, digested and macrophages isolated (defined by the gating strategy CD45+CD3−CD11b^hiF4/80+) and then using flow cytometric analysis, expressed as number of cells per gram of liver tissue (n = 3–4). **d** Serum TNF-α concentration from WT (Cre negative) and CLEC-2 deficient mice after 24 h after toxic liver injury injection (n = 6). Pre-treating the CLEC-2 deficient (CLEC1b^fl/flPf4Cre) mice with a TNF-α blocking antibody (etanercept) **e** restored liver injury in CLEC-2 deficient mice to WT levels and **f** reduced hepatic neutrophil infiltration after APAP injection (n = 4–8). Differences between groups assessed using a one-way ANOVA (Tukey's multiple comparisons test or Holm-Sidak) and unpaired Students t test or a Mann–Whitney test *P < 0.05, **P < 0.01, ***P < 0.001. Each data point is representative of an independent animal and data is presented as mean ± SEM.

challenge of managing patients who present with significant acute liver injuries, there are therapeutic implications for this work and inhibition of interactions between CLEC-2 and podoplanin are an exciting target to explore in human ALF.

## Methods
**Mice**. C57BL/6J mice were obtained from Harlan OLAC Ltd or from in-house colonies. Vav1-iCre+pdpn^fl/fl mice[23] and Pf4CreCLEC1b^fl/fl mice are described elsewhere[26]. All strains of genetically-altered mice were on a C57BL/6J background. Littermates that were negative for Cre recombinase and thus matched by genetic background, age and sex were used as controls. All mice were housed at the Biomedical Services Unit, University of Birmingham. All experiments were performed in accordance with UK laws (Animal [Scientific Procedures] Act 1986) with approval of local ethics committee and UK Home Office approval under PPL P2DF9DB6E and 70/7933 granted to the University of Birmingham.

**Human tissue**. All tissue used in this investigation was collected at the Liver and Hepatobiliary Unit, Queen Elizabeth Hospital, Birmingham, with prior written informed patient consent and local research ethics committee approval (06/Q702/61) Normal liver tissue was from hepatic resection margins from colorectal cancer metastases and donor tissue that was surplus to requirement for transplantation. Diseased liver tissue was obtained from explanted livers with APAP induced fulminant liver failure, obtained from patients in the Birmingham liver transplant programme.

**Induction of liver injury and therapeutic intervention**. Acute hepatic inflammation was induced using intraperitoneal (IP) injections of CCl₄ or APAP (both Sigma-Aldrich). CCl₄ was diluted 1:4 with mineral oil and injected intraperitoneally into mice at a dose of 1 ml/kg (control animals were treated with IP mineral oil alone). APAP was dissolved in phosphate buffered saline (PBS) at a temperature of 60 °C. The solution was cooled to 37 °C prior to injection at a final concentration of 350 mg/kg and control mice received IP PBS alone. Where indicated, mice were pre-treated intravenously with 5 µg/kg of functional grade

purified anti-podoplanin (Clone 8.1.1), or the corresponding isotype matched control antibody (golden Syrian hamster IgG) 24 h prior to being given either IP APAP or CCl$_4$. Experiments requiring TNF-α blockade were conducted using Etanercept (10 mg/kg, Amgen), control animals were given PBS alone. Mice were culled and tissue (liver and blood) collected either 6, 24 or 48 h after APAP injection and 24, 48 or 72 h after CCl$_4$ injection.

**Quantification of liver-infiltrating immune cells.** Mouse livers were harvested after the animal was euthanized under deep sedation (using isoflurane) and blood was collected by cardiac puncture. After weighing, a single liver lobe was dissociated in a gentleMACS homogenizer (Miltenyi Biotec). The homogenate was filtered through 70 μm filters (Falcon) and centrifuged at 1000 g to allow the cells to form a pellet. Following resuspension of the cells in RPMI 1640 medium (Roswell Park Memorial Institute) (Thermo-Fisher Scientific), a density gradient medium (OptiPrep™-Sigma Aldrich) was used to selectively isolate leucocytes. These immune cells were next stained with fluorophores as per manufacturers instructions and then analzyed flow cytometrically[64]. CD45$^+$ cells were gated (anti-CD45-PerCP-Cy5.5, clone 30-F11; BD Biosciences), and nonviable cells were excluded using a Zombie NIR™ Fixable Viability kit (Biolegend). Lymphocytes were characterized based on staining using a cocktail of anti–CD3-Pacific blue (clone 500A2); anti–CD4-PE (clone RM4-5); anti–CD8a-APC (clone 53-6.7); anti–CD19-APC-Cy7 or anti–CD19-BV510 (both clone 1D3); and anti–NK1.1-FITC (clone PK136) or DX5-FITC (clone DX5) antibodies (all from BD Biosciences) while the monocyte subsets were identified by staining with anti–CD11b-PE (clone M1/70; BD Biosciences); anti–GR1-APC (clone RB6-8C5; BD Biosciences); and anti–F4/80-FITC (clone BM8; eBioscience) antibodies. Podoplanin staining was identified using anti-podoplanin PE (clone 8.1.1; Biolegend). For intracellular TNF-α staining (, stained cells prepared as described above were suspended in intracellular buffer for 1 h and then permeabilized as per manufacturer's instructions (eBioscience) prior to staining with the anti-TNF-α antibody (anti- TNF-α PE-Cy7; Clone MP6-XT22; Biolegend). Absolute cell counts were determined with AccuCheck Counting Beads (Invitrogen), and the number of cells was normalized to the liver weight. Data were analysed using a CyAn ADP flow cytometer (Beckman Coulter) and FlowJo version 8.7 (Treestar software).

**Immunohistology and confocal microscopy.** Murine and human liver was snap-frozen in liquid nitrogen or fixed in 4% formaldehyde (NHS supplies) immediately upon removal. Paraffin-embedded 3 μm mouse and human tissue sections were then stained with either haematoxylin and eosin according to standard protocols or anti-podoplanin (mouse- eBio8.1.1, eBioscience; human- LpMab-21, Biolegend), anti-neutrophil elastase antibody (ab68672, Abcam) or anti-CYP2E1 (Biolegend) antibodies; these were then detected using horse-radish peroxidase conjugated secondary antibodies and developed using alkaline-phosphatase (ABComplex, Vector Laboratories) and 3,3′-diaminobenzidine tetrahydrochloride (DAB). Frozen tissue sections (mice and human) were prepared at a 7 μm thickness and then stained by immunohistochemistry (IHC) in Tris buffer (pH 7.6, Sigma) with 1% FCS (fetal calf serum)[21] to either detect murine podoplanin (eBio8.1.1, eBioscience); the secondary used was a goat anti-Syrian hamster Cy3 IgG (ab102318, Abcam) or human podoplanin (LpMab-21, Biolegend) the secondary in this case was a goat anti-mouse Cy3 IgG (ab97035, Abcam). Murine macrophages (FITC anti-mouse F4/80, BM8, Biolegend), human macrophages (FITC antihuman CD68, Y1/82 A, Biolegend), murine platelets (Brilliant Violet 605™ anti-mouse CD41, MWReg30, Biolegend) and human platelets (Alexa Fluor® 647 anti-human CD41/CD61 Antibody, PAC-1, Biolegend) were detected in frozen sections using the above described conjugated antibodies. Finally, mouse P-selectin was detected in frozen sections using a rat anti-mouse P-selectin antibody (mouse P-Selectin/CD62P, 127933, R&D systems); secondary used here was a goat anti-rat Cy3 IgG (ab98416, Abcam). Neutrophils were detected in paraffin sections using a neutrophil elastase AB (ab68672, Abcam). Manufacturer recommended control antibodies were used for each antibody. Primary and secondary antibodies were incubated for 60 min each at room temperature for paraffin sections. Primary and secondary antibodies on paraffin sections were used at concentrations of 1 μg/ml and 0.5 μg/ml respectively. All imaging was done at room temperature. DAB slides were mounted in (dityrene plasticizer xylene) DPX and images acquired at ×10, ×20 or ×40 magnification using a Leica CTR6000 microscope (Leica, Milton Keynes, UK) with QCapture software. Frozen sections were incubated with primary or conjugated antibodies for 90 min and secondary antibodies for 45 min, the sections were then counterstained with DAPI to highlight nuclei (10 μg/ml). Slides were mounted using Prolong Gold Anti-fade Reagent (Invitrogen, Paisley, UK) and images were captured at ×10, ×40 or ×63 magnification on a LSM510 laser scanning confocal microscope with a Zeiss AxioVert 100 M or a Zeiss LSM880 Airyscan detector set to SR mode using Zeiss LSM image software. Images were analysed using Zen blue (2012) software. Isotype matched control staining is shown in Supplementary Fig. 9.

Low magnification images brightfield and fluorescent images were acquired using a Carl Zeiss AxioScan Z1 Slide Scanner using a 3CCD colour 2MP Hitachi 1200×1600 HV-F202SCL camera. Images were analysed using Zen blue (2012) slide scan software. Where quantification was sought image J version 1.x (National Institutes of Health and the Laboratory for Optical and Computational Instrumentation, University of Wisconsin) was used.

**Biochemical assessment of liver injury.** Serum from mice was used for quantification of liver-specific enzymes using standard protocols on a Beckman Coulter AU400 analyser.

**Macrophage isolation.** Liver macrophages were isolated from murine livers using Blomhoff's method of selective plastic adherence[72]. Liver tissue was non-enzymatically digested in a gentleMACS homogenizer and cell pellets were re-suspended with 10 ml RPMI 1640 and centrifuged at 300×g for 5 min at 4 °C. After repeated slow spins at 300×g, the cell pellet which now mainly contained non-parenchymal liver cells including Kupffer cells and sinusoidal endothelial cells was then seeded into 6-well plate at a density of 1–3 × 10$^7$/well in Dulbecco's Modified Eagle's Medium (DMEM, Hyclone, USA) supplemented with 10% fetal bovine serum (FBS, Hyclone, USA) and 100U/ml Penicillin/Streptomycin (Sigma, USA) and incubated for 2 h in a 5% $CO_2$ atmosphere at 37 °C. Non-adherent cells were then removed from the dish by gently washing with PBS, leaving behind the adherent macrophages (Supplementary Fig. 1a).

**Platelet isolation.** Platelets were isolated from murine whole blood from either wild type or Pf4creCLEC1b$^{fl/fl}$ mice[24]. Briefly, a 25 gauge 1 ml syringe pre-filled with 100 μl of acid citrate dextrose was used to obtain murine blood via cardiac puncture. The whole blood was then centrifuged at 200×g for 7 min. The top layer of platelet rich plasma (PRP) was removed. In all, 200 μl of Tyrode's buffer composition was added to the remaining blood, and the suspension centrifuged for a further 5 min at 200×g, the top phase was again removed and added to the PRP from the previous step. To prevent platelet activation 1 μl of prostacyclin (1 mg/ml) was added per ml of PRP.

**Neutrophil activation.** The phagocytic ability of neutrophils was assessed using a phago-test™ kit as per manufacturer's instructions (BD Biosciences). Briefly, whole blood was extracted from deeply anesthetized mice (either CLEC-2 deficient or control mice), 24 h after APAP administration and placed into heparin containing tubes. The heparinized blood was then incubated with FITC labelled bacteria, the reaction was stopped after a pre-determined time by placing the solution on ice. Bacteria that were merely attached to the surface of the phagocyte rather than actually phagocytosed were washed off using a quenching solution. After erythrocyte lysis, neutrophils were flow-cytometrically isolated with the FITC$^+$ bacteria containing percentage identified as per the gating strategy described by the manufacturer and shown in Supplementary Fig. 1.

**Macrophage and platelet co-culture.** Macrophages isolated from wild type murine livers as described above, were stimulated with LPS (lipopolysaccharide-100 ng/ml, Sigma) and then co-cultured with either wild type or CLEC-2 deficient platelets. The supernatant was collected after 24 h and stored at −80 °C until use. The remaining cells were fixed in formalin and stained using conjugated fluorescent antibodies as describe above.

**TNF-α ELISA.** TNF-alpha levels were determined in either mouse serum or cell culture supernatant from the macrophage assays described above was determined using a sandwich ELISA technique according to manufacturer's instructions (eBioscience).

**Quantitation of podoplanin mRNA from human liver.** RNA was extracted from human livers using a RNEasy kit (Qiagen, UK) and cDNA was generated using a High capacity cDNA reverse transcription kit (ThermoFisher, UK) according to manufacturer's instructions. Real time PCR (pdpn assay ID Hs00366766_m1) was carried out on each sample in triplicate using a Roche Lightcycler 480 machine Taqman Assay Mix (Life Technologies, UK). Cycling conditions were: 95 °C 10 s, 60 °C 50 s, 72 °C 1 s. For relative quantification analysis data were normalised to housekeeping gene(s) (GADPH; assay ID Hs02786624_g1) using 'E-analysis' (Roche Diagnostics Ltd) and the device software.

**Statistical analysis.** Data is presented as mean ± standard error of the mean (SEM). Numbers of animals in each model are stated in the figure legends. Differences were analysed using GraphPad Prism software (GraphPad software Inc., La Jolla, CA, USA). Normality was checked using either the Shapiro-Wilk or Kolmogorov–Smirnov test (depending upon the $n$ number in that particular experiment). Normally distributed data were compared using the Students unpaired $t$-test and non-normally distributed data were compared using the Mann–Whitney test. When more than two groups were being compared an ordinary one-way ANOVA was used. $P$ values of ≤0.05 were considered significant.

## Data availability
Data available from authors upon reasonable request. The source data underlying Figs. 1d–e, 2a, b, e, 3a, 4b, d–g, 5a–f, Supplementary Figs. 6, 7b and 8b are provided as a Source Data File.

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

## Author contributions

A.C. wrote the grant, designed the research studies, conducted experiments, acquired data, analysed data and wrote the manuscript. G.J.W., M.H. and L.S: designed and conducted experiments. G.J.W. and M.H. critiqued paper. D.P., E.S., D.H., R.S. and S.B. conducted experiments. C.J.W. designed experiments. S.W. provided reagents and assisted with conducting experiments. P.L., S.P.W. and D.H.A. wrote the grant, designed the research studies and wrote the paper.

## Competing interests

The authors declare no competing interests.
