## [Peer Review File · Nature Communications]

Reviewers' Comments:

Reviewer #2:

Remarks to the Author:

In mouse models of acetaminophen- and CCl₄-induced liver injury, the authors present evidence that the platelet receptor CLEC-2 contributes to worsening of injury by interacting with its ligand, podoplanin, perhaps on macrophages. Pharmacologic or genetic reduction of the CLEC-2-podoplanin interaction leads to amelioration of liver injury, and reduction of TNF or neutrophils prevents that amelioration, suggesting a role for these mediators in injury repair (or in halting injury progression). The results are of interest, and the manuscript is well written overall. There are several issues as noted below that require attention and must be addressed. There are some results that lack rigor as presented; deletion of these would strengthen the work.

Title and abstract: Much of the work described relates to CCl₄ as well as acetaminophen, yet CCl₄-induced injury is not mentioned in the title or the abstract. It should be.

P 5, para 2, reference to the role of neutrophils in APAP-induced liver injury would be appropriate to this work.

Introduction and Discussion: The idea that platelets contribute to liver injury from APAP is not novel. The authors fail to mention or discuss published works in the mouse model that implicate platelets in the pathogenesis of liver injury from APAP. This work is highly germane to the manuscript.

Fig. 1: In Fig. 1a, the podoplanin staining in PSC, NASH and ALD looks predominately vascular and not associated with macrophages. In the other conditions, it is hard to tell with what cells the staining is associated. Thus, the nature of the staining in these conditions does not necessarily support association with macrophages. More relevant to this study would be to show staining in livers from APAP-poisoned patients. Overall, Fig 1a lacks rigor and should be deleted. It is not needed to make points about APAP and CCl₄ toxicity in the mouse.

Fig. 1b: Where is the lobular location of the staining in the toxicant-treated mice? It is hard to tell whether the staining is periportal or pericentral; the location could have implications for interpretation. For example, periportal regions are spared during CCl₄ or APAP injury, so why would macrophages and platelets be activated there?

Fig 1c: The claimed podoplanin costaining of macrophages is unconvincing as presented. In Fig. 1e, the merged panel clearly shows costaining, but Fig. 1c lacks this rigor.

I suggest deleting all panels but Fig. 1c, which makes a clear point that is most germane to the rest of the study.

Fig2: Miyakawa and coworkers (Blood 126: 1835, 2015) showed that almost complete depletion of platelets in APAP-treated mice led to a decrease in ALT of about 50%. In light of this result, it seems remarkable that CLEC-2 deficiency should have the profound effect on ALT depicted in Fig 2a. The authors' thoughts on this difference should be discussed. They also identified protease-activated receptor 4 as important in the platelet response; how does PAR-4 relate to CLEC-2? Also, on p 9, reference is made to histological staining at 24hr in APAP-treated mice, but Fig 2b indicates 48 hr; which is correct?

Fig 4d: The label on the axis should be in a larger font.

Fig 5c: Did CLEC-2 deficiency lead to a statistically significant increase in neutrophils in liver in this study? This is not indicated in the figure.

Discussion:

P 20, para 1: "We report a novel mechanism through which platelets accelerate recovery from acute liver injury in response to APAP and CCl4 toxicity." Is this what the authors mean, or should it be "platelets retard recovery...."?

Is the CLEC-2 deficiency in platelets really accelerating recovery (eg, replicative repair) of liver or is it diminishing progression of injury? 24hr after APAP seems very early for complete repair by hepatocyte proliferation to occur.

P 22: TNF can have a damaging role in liver injury as well as contribute to recovery from injury. The authors should acknowledge this. In other models such as LPS-induced liver injury, TNF has a damaging role.

Reviewer #3:

Remarks to the Author:

Chauhan and colleagues provide an interesting study on the role of platelets in sterile liver injury and its consequences for liver failure and regeneration. They show that CLEC-2 dependent platelet activation reduces liver injury, which seems to be dependent on TNF- α mediated neutrophil recruitment. The authors offer a novel treatment concept since absence of CLEC-2 is not associated with alterations in major hemostatic pathways.

The study is mostly well conducted and contains novel data from transgenic mouse models and some human data. To this reviewer, the beneficial role of blocking CLEC-2 dependent platelet activation in liver failure is unequivocally demonstrated. However, the underlying mechanisms have only been clarified partially. Specifically, the precise role of increased TNF- α signaling and boosted neutrophil recruitment in this setting remains somewhat enigmatic. Further, the potentially beneficial role of increased neutrophil influx into the acutely inflamed liver is in contrast to most of the published literature. Thus, a further workup should clarify this apparent contradiction.

Specific comments:

- Authors demonstrate an association between platelets and podoplanin-positive macrophages (figure 1). Is this association dependent on Clec-2 expression on platelets?
- Figure 1e: CD41 fluorescence staining perfectly matches the F4/80 staining, suggesting the close association of platelets with macrophages. Please show control stainings for each fluorochrome to exclude unspecific staining and spectral overlap.
- Since, podoplanin is highly upregulated in the endothelium in inflammatory conditions, what is the role of the vessel wall (also discussed by the authors, and shown in chronic human liver disease), as opposed to macrophage podoplanin, in this setting? Vav-Cre has been shown to recombine efficiently also in the endothelial lineage (Joseph CellStemCell 2013, Georgiades Genesis 2002) and, therefore, conditional deletion in vavCre transgenic mice is not exclusive to the hematopoietic lineage.
- Indeed, Pf4CreCLEC1bfl/f mice have deletion of CLEC-2 gene (CLEC1b) in cells of the megakaryocytic lineage. However, PF4 is also highly abundant in macrophages. Is Clec-2 expressed in the macrophage lineage? And if so, does Clec-2 deletion in macrophages have any biological consequences in general, and in the here described model specifically?
- Authors demonstrate that abrogating CLEC-2-dependent platelet activation enhances recovery from liver injury by increasing hepatic neutrophil infiltration (in a TNF- α dependent manner). This

statement suggests that neutrophil infiltration per se is beneficial in acetaminophen-induced liver injury (APAP). However, a large body of work has previously indicated a great relevance of neutrophils in the pathogenesis of this condition, with neutrophil depletion having beneficial effects on mouse outcome (Lawson ToxicolSci 2000, Ishida EurJImmunol 2006, Liu Hepatology 2006, Marques Hepatology 2012). Can the authors clarify this apparent contradiction? Are there differences in the activation status of neutrophils in wildtype versus Clec-2 deficient mice? Are there differences in the presence of hepatoprotective cytokines (refers to REF #34, Graubardt FrontImmunol 2017).

- In the abstract, authors conclude that "These results provide the first demonstration that platelets actively participate in the hepatic sterile inflammatory response". I suggest to revise this conclusions, since some studies have shown the contribution of platelets to sterile liver injury (e.g. REF #18, Miyakawa Bloo 2015, Sullivan ToxicolSci 2010).

- Statistical analysis: (1) It should be stated in the figure legends whether SEM or SD was used. (2) SD should be shown, instead of SEM, for low numbers of biological repeats. (3) Authors show mean plus s.e.m. (not +/-).

- Please omit "data not shown".

- Abstract: this can (add "be") manipulated for therapeutic benefit

Reviewer #4:

Remarks to the Author:

Platelet CLEC2 and podoplanin are critical in several biological processes, including blood/lymphatic vessel separation, vascular integrity, thrombosis, and lung development. The authors revealed a new role of platelet CLEC2 signaling in toxic liver injury. Their results show that blocking CLEC-2 enhanced liver recovery from toxic injury by increasing TNF-dependent neutrophil recruitment. However, their conclusion is not supported by convincing evidence. In addition, there are no mechanistic insights into how platelet CLEC-2 increases TNF and how infiltrated neutrophils ameliorate liver injuries.

Major comments:

1. The title of the paper focuses on sterile acetaminophen-induced liver injury. However, the study uses liver samples from different human diseases, from different mouse liver injury models, and even includes LPS-induced inflammation. Unless the authors provide evidence that these different types of liver injury share the same pathology, the manuscript should focus on acetaminophen-induced liver injury.

2. A main finding is that podoplanin is upregulated in the Kupffer cells during liver injury. However, results in Fig. 1 supporting this finding are of poor quality: 1) human or mouse samples from different pathologies were used; 2) the resolution of most images are poor; and 3) the staining pattern of podoplanin in most images does not seem to be in the Kupffer cells. Some appear to be in endothelial cells (lymphatics?, Fig. 1a, b), and some look like in infiltrated macrophages but not Kupffer cells (Fig. 1c, e).

3. The study lacks mechanistic insights. No evidence shows platelets were activated by CLEC-2 and podoplanin interaction during toxic liver injury in mice nor how this may lead to increased TNF in Kupffer cells. How infiltrated neutrophils play protective roles in the recovery of APAP-induced liver injury is not studied. Various types of cells can release TNF during inflammation, including macrophage, platelets, and endothelial cells, etc. This manuscript provides no evidence that TNF is specifically released from the Kupffer cell.

4. Most experiments in Figs. 1 - 3 lack untreated groups from WT and KO mice as control. Many labels in Figs. are too small to read.

Minor comments

5. Quantitative data should be provided for results, such as in Fig. 1a, e.
6. The "normalized to GAPDH" in Fig. 1d is inconsistent with the legend description of "normalized to 18S".
7. Fig. 1c, boxed right panels do not seem to be from the middle panels. "F480" label in the middle lower panel is wrong.
8. There are many broken sentences, for example, page 18, paragraph 1, line 8 and line 14; page 20, paragraph 1, line 3; etc.

CLGR
CENTRE FOR LIVER AND
GASTROINTESTINAL RESEARCH

UNIVERSITY OF
BIRMINGHAM

Centre for Liver and Gastroenterology Research
Institute for Immunology and Immunotherapy
Institute for Biomedical Research
The Medical School
University of Birmingham
Birmingham B15 2TT
UK

19th November 2018

Re : Response to referees critique NCOMMS-18-16530

Dear Sirs,

Please find enclosed our detailed responses to the individual comments from the referees regarding our above detailed manuscript. The reviewer comments are in italics and our responses in bold.

Reviewer 2

This referee noted that our results were of interest and that the manuscript was well written overall. They added 'In mouse models of acetaminophen- and CCl₄-induced liver injury, the authors present evidence that the platelet receptor CLEC-2 contributes to worsening of injury by interacting with its ligand, podoplanin, perhaps on macrophages. Pharmacologic or genetic reduction of the CLEC-2-podoplanin interaction leads to amelioration of liver injury, and reduction of TNF or neutrophils prevents that amelioration, suggesting a role for these mediators in injury repair (or in halting injury progression). The results are of interest, and the manuscript is well written overall. There are several issues as noted below that require attention and must be addressed. There are some results that lack rigor as presented; deletion of these would strengthen the work. Title and abstract: Much of the work described relates to CCl₄ as well as acetaminophen, yet CCl₄-induced injury is not mentioned in the title or the abstract. It should be.

We have amended the title to a more generic reference to 'toxic' liver injury, have added CLEC-2 mediated platelet activation to the title and have also added CCl₄ to the abstract. We have also removed some of the less robust data from human chronically injured livers (Fig 1a) in reference to this referees comment here and below.

'P 5, para 2, reference to the role of neutrophils in APAP-induced liver injury would be appropriate for this work' and also' Introduction and Discussion: The idea that platelets contribute to liver injury from APAP is not novel. The authors fail to mention or discuss published works in the mouse model that implicate platelets in the pathogenesis of liver injury from APAP. This work is highly germane to the manuscript.'

Here we have added references from Kubes.,et al, Slaba et al., Lam et al., and Miyakawa et al., as requested. These detail the previously described roles of platelets in APAP induced liver injury.

'Fig. 1a, the podoplanin staining in PSC, NASH and ALD looks predominately vascular and not associated with macrophages. In the other conditions, it is hard to tell with what cells the staining is associated. Thus, the nature

of the staining in these conditions does not necessarily support association with macrophages. More relevant to this study would be to show staining in livers from APAP-poisoned patients. Overall, Fig 1a lacks rigor and should be deleted. It is not needed to make points about APAP and CCl₄ toxicity in the mouse'

As suggested by the reviewer we have removed Fig 1a as this pertains to podoplanin expression in chronically injured fibrotic liver injury; this is distinct to acute toxic liver injury as exemplified by APAP poisoning which is the focus of our paper. Furthermore, again as suggested by the reviewer we have included high quality staining from other APAP poisoned patients to not only clearly demonstrate that macrophages upregulate podoplanin but also clarify that the location starts pericentrally-where the necrotic injury due to APAP/CCl₄ occurs (shown in 1d lower row) and then affects the periportal areas (1d upper row).

'Fig. 1b: Where is the lobular location of the staining in the toxicant-treated mice? It is hard to tell whether the staining is periportal or pericentral; the location could have implications for interpretation. For example, periportal regions are spared during CCl₄ or APAP injury, so why would macrophages and platelets be activated there?'

This is an important consideration. We were remiss in not showing images that allow for lobular localization of cells to be clearly defined. We have added new images to Figure 1 show that the upregulation begins in pericentral areas (where toxic injury due to APAP/CCl₄ begins-1d lower row-pericentral) and then it is likely that that cells recruited and activated in the sinusoids exit the liver at the portal areas (thus 1d upper row shows peri portal macrophages expressing podoplanin) and have amended the main body text accordingly. Furthermore the new figure 2c (top row) clearly shows pericentral platelet activation and sequestration to macrophages in mice. Our new data next demonstrate that by removing CLEC-2 we are able to reduce both platelet activation as gauged by P-selectin expression (median % area expression WT vs CLEC-2 deficient 2.75 vs 0.577; n=8 random fields-supplementary 7) and attachment to macrophages in areas of maximal injury (ie pericentrally- figure 2c-bottom row, and graph 2d). We have further amended our discussion to explain this.

'Fig 1c: The claimed podoplanin costaining of macrophages is unconvincing as presented. In Fig. 1e, the merged panel clearly shows costaining, but Fig. 1c lacks this rigor. I suggest deleting all panels but Fig. 1c, which makes a clear point that is most germane to the rest of the study'.

We have added new high definition images using airy scanning to both Fig 1 (1d, 1e) and Fig 2 (2c) with the corresponding isotype matched controls (supplementary 6) to improve the rigour of our statements. These clearly demonstrate pericentral podoplanin upregulation, as well as distinct cellular upregulation of podoplanin upon macrophages after acute liver injury in both mice and humans.

'Fig2: Miyakawa and coworkers (Blood 126: 1835, 2015) showed that almost complete depletion of platelets in APAP-treated mice led to a decrease in ALT of about 50%. In light of this result, it seems remarkable that CLEC-2 deficiency should have the profound effect on ALT depicted in Fig 2a. The authors' thoughts on this difference should be discussed. They also identified protease-activated receptor 4 as important in the platelet response; how does PAR-4 relate to CLEC-2?'

We have demonstrated that CLEC-2 deficiency results in about a tenfold reduction in ALT 48 hours following CCl₄ administration or 24 hours following paracetamol exposure but it important to note that in the early period following administration of either agent, transaminase levels were similar to wild type mice. Thus at 6 hours post paracetamol, mean ALT was 1521IU/l in WT and 1194IU/L in CLEC-2 deficient mice. This is about a 20% reduction. The data in the Miyakawa study (Fig 4) also looked around 6 hours after administration and showed higher ALT (around 2 x our values) and a 40% reduction if platelets were depleted using CD41 antibody. We utilized a dose paracetamol similar to that described by Miyakawa et al., but did not fast our animals prior to administration. Thus it is not surprising that their transaminase response suggested a more significant injury. However it is clear that the magnitude of the effect of platelet inhibition/reduction depends upon the timing of measurement with a profound alteration during the resolution phase and gradual return to baseline and so our results are not dissimilar at the early timepoint. We have not

specifically addressed the role of platelet activation via PAR-4 in our study, however the Miyakawa study actually shows that whilst platelet deficiency did reduce liver damage, platelet PAR-4 was not contributory to this protection as mice that deficient in platelet PAR-4 (the authors used chimeric mice with a selective (PAR)-4 deficiency on hematopoietic cells) exhibited no protection from APAP toxicity. This suggests that alternative pathways of platelet activation are relevant in APAP toxicity; this lends further support to our data as we show that it is the CLEC-2/podoplanin axis that is key.

'Also, on p 9, reference is made to histological staining at 24hr in APAP-treated mice, but Fig 2b indicates 48 hr; which is correct? Fig 4d: The label on the axis should be in a larger font. Fig 5c: Did CLEC-2 deficiency lead to a statistically significant increase in neutrophils in liver in this study? This is not indicated in the figure. Fig 4d: The label on the axis should be in a larger font. Fig 5c: Did CLEC-2 deficiency lead to a statistically significant increase in neutrophils in liver in this study? This is not indicated in the figure.'

Our apologies for any confusion or lack of clarity here – the staining for podoplanin is indeed shown 24 hours post APAP overdose in Figure 1 whilst the biochemical and histological scoring in figure 2 ranges from 6 to 48 hours. We have clarified this in the final manuscript. We have also adjusted figure axes size for consistency and clarity. We have also added statistical notation to Figure 5C to show that the increase in neutrophil numbers was indeed significant in CLEC-2 deficient mice.

'Discussion : P 20, para 1: "We report a novel mechanism through which platelets accelerate recovery from acute liver injury in response to APAP and CCl4 toxicity." Is this what the authors mean, or should it be "platelets retard recovery...."?'

The reviewer is correct here – we have amended the text accordingly

'Is the CLEC-2 deficiency in platelets really accelerating recovery (eg, replicative repair) of liver or is it diminishing progression of injury? 24hr after APAP seems very early for complete repair by hepatocyte proliferation to occur.'

This is a fair point and both possibilities could occur. Our transaminase values are similar in both WT and knockout mice at early timepoints suggesting that the injury occurs as normal. We have performed further analyses of hepatocyte replication via Ki67 staining of the tissue (supplementary 7) and whilst there is slightly more Ki67 activity in the CLEC-2 deficient mice suggesting greater regeneration, this response is not statistically viable (2.3 vs 2.9 Ki67 positive hepatocytes per randomly selected central field; n=10 fields, median shown, P=0.7416). We feel the lower serum transaminase activity in the CLEC-2 deficient mice is thus due to greater neutrophil infiltration and as has been described previously^{1,2}, neutrophil mediated clearance of dying/necrotic hepatocytes and not regeneration per se.

'P 22: TNF can have a damaging role in liver injury as well as contribute to recovery from injury. The authors should acknowledge this. In other models such as LPS-induced liver injury, TNF has a damaging role'

Yes this is absolutely correct, we have amended our narrative text to highlight the pleiotropic effects of TNF in the discussion and have added new references to cite the appropriate evidence.

Reviewer 3

Chauhan and colleagues provide an interesting study on the role of platelets in sterile liver injury and its consequences for liver failure and regeneration. They show that CLEC-2 dependent platelet activation reduces liver injury, which seems to be dependent on TNF- α mediated neutrophil recruitment. The authors offer a novel treatment concept since absence of CLEC-2 is not associated with alterations in major hemostatic pathways. The study is mostly well conducted and contains novel data from transgenic mouse models and some human data. To this reviewer, the beneficial role of blocking CLEC-2 dependent platelet activation in liver failure is unequivocally demonstrated. However, the underlying mechanisms have only been clarified partially. Specifically, the precise role of increased TNF- α signaling and boosted neutrophil recruitment in this setting

remains somewhat enigmatic. Further, the potentially beneficial role of increased neutrophil influx into the acutely inflamed liver is in contrast to most of the published literature. Thus, a further workup should clarify this apparent contradiction.

We were pleased that this reviewer found our work of interest and that they felt that ‘the beneficial role of blocking CLEC-2 dependent platelet activation in liver failure is unequivocally demonstrated’. However in common with the other reviewers they requested clarification as to mechanism, so we have addressed their specific comments as follows:

‘Authors demonstrate an association between platelets and podoplanin-positive macrophages (figure 1). Is this association dependent on CLEC-2 expression on platelets?’

Whilst it is likely that the association between platelets and macrophages is mediated by multiple receptor pairings including but not limited to Gp1b:mac-1 and CD40:CD40L interactions, we have now added additional data to support a key role for direct association between CLEC-2 expressing platelets and podoplanin expressing macrophages (Figure 1e). We next demonstrate with fluorescent microscopic pictures (the new figure 2c) that F480+ macrophages interact with platelets, and that this results in activation of platelets as gauged by expression of P-selectin during acute liver injury; furthermore we demonstrate that this activation is diminished in CLEC-2-deficient animals undergoing the same acute liver injury (Fig 2c-d and supplementary 7) We have added text to clarify this in the amended manuscript. This new data also assists with response to the specific query below :

‘Figure 1e: CD41 fluorescence staining perfectly matches the F4/80 staining, suggesting the close association of platelets with macrophages. Please show control stainings for each fluorochrome to exclude unspecific staining and spectral overlap.’

We have now added new images and isotype control stainings to address this comment.

‘Since, podoplanin is highly upregulated in the endothelium in inflammatory conditions, what is the role of the vessel wall (also discussed by the authors, and shown in chronic human liver disease), as opposed to macrophage podoplanin, in this setting? Vav-Cre has been shown to recombine efficiently also in the endothelial lineage (Joseph CellStemCell 2013, Georgiades Genesis 2002) and, therefore, conditional deletion in vavCre transgenic mice is not exclusive to the hematopoietic lineage.’

The reviewer is correct and podoplanin is highly expressed on lymphatic endothelium³ in particular in homeostatic conditions and as we and others have noted (this data has now been removed as not of relevance in acute liver injuries) upregulated on sinusoidal endothelium in the context of chronic disease. Indeed we and others have shown that direct platelet:endothelial interactions are important for the recruitment of lymphocytes^{4 5,6} and in an acute setting this can contribute to situations such as ischaemic reperfusion injury⁷. However in the toxic injuries (CCI₄/APAP) we are reporting here, we observe minimal endothelial podoplanin expression in association with platelet sequestration (See Figure 1) thus we believe the macrophage:platelet interaction to be of more significance to our response.

Here there is indeed some evidence to suggest that in murine systems Vav is expressed in endothelial precursors but other studies suggest no non-haematopoietic⁸ expression and critically it is absent from adult murine liver⁹. Furthermore, our data (from wild type mice and humans) clearly indicates that in acute toxic liver injury podoplanin upregulation is confined to macrophages and thus any effects attributable to endothelial podoplanin expression would negligible in acute toxic liver injuries anyway. There are no reports of expression of CLEC-2 on non-lymphatic hepatic endothelial cells. Thus we have added text to our discussion section to clarify this situation.

'Indeed, Pf4CreCLEC1bfl/f mice have deletion of CLEC-2 gene (CLEC1b) in cells of the megakaryocytic lineage. However, PF4 is also highly abundant in macrophages. Is CLEC-2 expressed in the macrophage lineage? And if so, does Clec-2 deletion in macrophages have any biological consequences in general, and in the here described model specifically?'

No CLEC-2 is not expressed upon macrophages, this has been shown previously¹⁰, although circulating monocyte populations do express CLEC-2 in some cases (see below).

'Authors demonstrate that abrogating CLEC-2-dependent platelet activation enhances recovery from liver injury by increasing hepatic neutrophil infiltration (in a TNF- α dependent manner). This statement suggests that neutrophil infiltration per se is beneficial in acetaminophen-induced liver injury (APAP). However, a large body of work has previously indicated a great relevance of neutrophils in the pathogenesis of this condition, with neutrophil depletion having beneficial effects on mouse outcome (Lawson ToxicolSci 2000, Ishida EurJImmunol 2006, Liu Hepatology 2006, Marques Hepatology 2012). Can the authors clarify this apparent contradiction?'

We agree that the role of neutrophils in acute liver injuries is complex and context dependent. However whilst in certain liver pathologies a proven damaging effect of neutrophils is observed, the strategy used to deplete neutrophils (timing of depletion and choice of agent (eg GR-1 vs ICAM-1 or chemokine blockade) is crucial. Most studies showing neutrophil depletion to be beneficial in acute liver injury depended on pre-treatment of mice with the neutrophil depleting antibody; the data was not reproducible on treating the mice post injury. Thus it was likely an off target preconditioning effect that resulted in the observed protective phenotype. Furthermore numerous authors have demonstrated that not only do neutrophils not drive APAP toxicity but rather, hepatic neutrophil accumulation as part of the sterile inflammatory response is key in driving liver recovery. We have modified our discussion to reflect this and have incorporated the key references in the altered manuscript.

'Are there differences in the activation status of neutrophils in wildtype versus CLEC-2 deficient mice? Are there differences in the presence of hepatoprotective cytokines (refers to REF #34, Graubardt FrontImmunol 2017).'

Extensive analysis of CLEC-2 deficient mice suggests that some peripheral blood myeloid populations express CLEC-2 on their surface and that this is lost upon migration into peripheral lymphoid organs. No changes in neutrophil activation status have been reported in these animals¹⁰.

'In the abstract, authors conclude that "These results provide the first demonstration that platelets actively participate in the hepatic sterile inflammatory response". I suggest to revise this conclusions, since some studies have shown the contribution of platelets to sterile liver injury (e.g. REF #18, Miyakawa Blood 2015, Sullivan ToxicolSci 2010).'

Agreed – we have modified this sentence

'Statistical analysis: (1) It should be stated in the figure legends whether SEM or SD was used. (2) SD should be shown, instead of SEM, for low numbers of biological repeats. (3) Authors show mean plus s.e.m. (not +/-). - Please omit "data not shown".'

We have corrected our figure legends and edited text accordingly; all figures now show standard deviation

Reviewer 4

'Platelet CLEC2 and podoplanin are critical in several biological processes, including blood/lymphatic vessel separation, vascular integrity, thrombosis, and lung development. The authors revealed a new role of platelet CLEC2 signaling in toxic liver injury. Their results show that blocking CLEC-2 enhanced liver recovery from toxic

injury by increasing TNF-dependent neutrophil recruitment. However, their conclusion is not supported by convincing evidence. In addition, there are no mechanistic insights into how platelet CLEC-2 increases TNF and how infiltrated neutrophils ameliorate liver injuries.

Major Comments : 1. The title of the paper focuses on sterile acetaminophen-induced liver injury. However, the study uses liver samples from different human diseases, from different mouse liver injury models, and even includes LPS-induced inflammation. Unless the authors provide evidence that these different types of liver injury share the same pathology, the manuscript should focus on acetaminophen-induced liver injury.'

We agree that our title/abstract did not capture the full breadth of our study and have amended it accordingly. We have also removed the chronic liver injury data as this was not relevant to this study; the models of in vivo mouse injury we have used are CCl₄ and APAP, we have thus amended the title and abstract to include toxic liver injuries.

'2. A main finding is that podoplanin is upregulated in the Kupffer cells during liver injury. However, results in Fig. 1 supporting this finding are of poor quality: 1) human or mouse samples from different pathologies were used; 2) the resolution of most images are poor; and 3) the staining pattern of podoplanin in most images does not seem to be in the Kupffer cells. Some appear to be in endothelial cells (lymphatics?, Fig. 1a, b), and some look like in infiltrated macrophages but not Kupffer cells (Fig. 1c, e).'

To address these and similar comments about figure quality and clarity from other referees we have added new imaging data and improved our descriptions in the manuscript legends and main body. We have further removed all images pertaining to chronic liver injury; this is where the 'endothelial staining' is noted.

'The study lacks mechanistic insights. No evidence shows platelets were activated by CLEC-2 and podoplanin interaction during toxic liver injury in mice nor how this may lead to increased TNF in Kupffer cells. How infiltrated neutrophils play protective roles in the recovery of APAP-induced liver injury is not studied. Various types of cells can release TNF during inflammation, including macrophage, platelets, and endothelial cells, etc. This manuscript provides no evidence that TNF is specifically released from the Kupffer cell. Most experiments in Figs. 1 - 3 lack untreated groups from WT and KO mice as control. Many labels in Figs. are too small to read. Quantitative data should be provided for results, such as in Fig. 1a.'

We have added new data which confirms that ligation of platelet CLEC-2 by podoplanin positive macrophages leads to expression of P-Selectin (Figure 1e, 2c-d, supplementary 7; % median area expression WT vs CLEC-2 deficient 2.75 vs 0.577; n=8 random fields). Whilst we agree with the referee in his assertion that TNF- α is produced by many cell types we would disagree when he suggests that we have not shown that our elevated TNF- α production is Kupffer cell-dependent. The data in figure 5 confirms that we see higher circulating TNF- α levels in CLEC-2 deficient animals [median WT-181.6ng/ml (n=5)vs CLEC-2 deficient-608.7ng/ml (n=6); P=.0002]and also that in coculture, we see more TNF- α production from macrophages exposed to CLEC-2 deficient platelets (TNF- α -WT (n=5) vs CLEC-2 deficient(n=4); 31.64 ng/ml vs 48.90ng/ml). Similarly we have shown quantitative PCR data for expression of podoplanin in Figure 1 (non-injured liver vs Injured liver; median PCR 0.0002330, n=4 vs 0.001525, n=4; P=0.0286) and transaminases in Figure 2 [(48 hour median ALT CCl₄ WT vs CLEC-2 deficient; 4759 IU/L (n=8) vs 305 IU/L (n=4) P=0.004; 24 hour median ALT APAP WT vs CLEC-2 deficient; 5906 IU/L(n=4) Vs 272.5IU/L(n=4)] to quantify extent of liver injury. The protective role of neutrophils has been discussed in our revised discussion and we have amended figure axes for improved readability.

The "normalized to GAPDH" in Fig. 1d is inconsistent with the legend description of "normalized to 18S". Fig. 1c, boxed right panels do not seem to be from the middle panels. "F480" label in the middle lower panel is wrong. There are many broken sentences, for example, page 18, paragraph 1, line 8 and line 14; page 20, paragraph 1, line 3; etc.'

This referee also noted inconsistencies and proofreading errors in our manuscript for which we apologise. These have been corrected in our amended manuscript.

We enclose the following documents with our re-submission-

- 1) Full manuscript with changes highlighted and figures embedded.
- 2) Full manuscript with no highlighting and figures embedded.
- 3) Full manuscript with no highlighting, no figures embedded, figure legends at the end.
- 4) Figures as a separate powerpoint file which contains all the original high-resolution images.
- 5) Figures as a standalone pdf.
- 6) All requested checklists

Please do not hesitate to contact me if you require additional information
Yours sincerely

Dr Patricia Lalor
p.f.lalor@bham.ac.uk

- 1 Williams, C. D., Bajt, M. L., Farhood, A. & Jaeschke, H. Acetaminophen-induced hepatic neutrophil accumulation and inflammatory liver injury in CD18-deficient mice. *Liver Int* **30**, 1280-1292, doi:10.1111/j.1478-3231.2010.02284.x (2010).
- 2 Williams, C. D. *et al.* Neutrophil activation during acetaminophen hepatotoxicity and repair in mice and humans. *Toxicol Appl Pharmacol* **275**, 122-133, doi:10.1016/j.taap.2014.01.004 (2014).
- 3 Hirakawa, S. *et al.* Identification of vascular lineage-specific genes by transcriptional profiling of isolated blood vascular and lymphatic endothelial cells. *Am.J.Pathol.* **162**, 575-586 (2003).
- 4 Iannacone, M. *et al.* Platelets mediate cytotoxic T lymphocyte-induced liver damage. *Nat.Med.* **11**, 1167-1169 (2005).
- 5 Lalor, P. & Nash, G. B. Adhesion of flowing leucocytes to immobilized platelets. *Br.J.Haematol.* **89**, 725-732 (1995).
- 6 Lalor, P. F., Herbert, J., Bicknell, R. & Adams, D. H. Hepatic sinusoidal endothelium avidly binds platelets in an integrin-dependent manner, leading to platelet and endothelial activation and leukocyte recruitment. *Am J Physiol Gastrointest Liver Physiol* **304**, G469-478, doi:10.1152/ajpgi.00407.2012 (2013).
- 7 Pak, S. *et al.* Platelet adhesion in the sinusoid caused hepatic injury by neutrophils after hepatic ischemia reperfusion. *Platelets* **21**, 282-288, doi:10.3109/09537101003637265 (2010).
- 8 Ogilvy, S. *et al.* Promoter elements of vav drive transgene expression in vivo throughout the hematopoietic compartment. *Blood* **94**, 1855-1863 (1999).
- 9 Bustelo, X. R., Rubin, S. D., Suen, K. L., Carrasco, D. & Barbacid, M. Developmental expression of the vav protooncogene. *Cell Growth Differ* **4**, 297-308 (1993).
- 10 Lowe, K. L. *et al.* The expression of mouse CLEC-2 on leucocyte subsets varies according to their anatomical location and inflammatory state. *Eur J Immunol* **45**, 2484-2493, doi:10.1002/eji.201445314 (2015).

Reviewers' Comments:

Reviewer #2:

Remarks to the Author:

The authors have substantially addressed my original concerns. They should consider the following in their revision:

Fig 1a: How long after the human acute APAP poisoning occurred were these livers taken for analysis? Was this a liver biopsy or a section taken after death?

It seems that the penultimate comment by reviewer 4 regarding Kupffer cells being the source of TNF in vivo has not been addressed. The results stated in the author response do not address whether Kupffer cells are the TNF source in the APAP mouse model. Interference with Kupffer cell production of TNF would be needed to address this. This is not needed for publication of this manuscript, but the authors should be careful about ascribing cause and effect in this regard.

Line 140: "human and murine livers respond to acute liver injuries by increasing hepatic podoplanin expression" This statement is too broad and should be confined to what the authors actually showed, ie, that podoplanin seems to be expressed in APAP overdose injury. Whether it occurs in other injuries in humans is not addressed.

Line 356: Is there a misplaced comma here (should be after "in vivo")?

Reviewer #3:

Remarks to the Author:

All issues raised by this reviewer have been sufficiently addressed.

Reviewer #4:

Remarks to the Author:

The authors made effort to revise the manuscript based on reviewers' comments; however, there were only partial revisions. Primary concerns remain. Specifically, 1) the staining of podoplanin in most images looks like in infiltrated macrophages but not Kupffer cells, 2) LPS was still used in some in vitro experiments, and 3) P-selectin is not a reliable maker for platelet activation in tissue sections, as it is constitutively stored within platelets. The revised manuscript also still lacks mechanistic insights. No evidence shows how deficiency of platelet CLEC-2 leads to increased TNF in Kupffer cells. How infiltrated neutrophils play protective roles in the recovery of APAP-induced liver injury is not studied. This manuscript provides no evidence that TNF is specifically released from the Kupffer cell.

CLGR
CENTRE FOR LIVER AND
GASTROINTESTINAL RESEARCH

UNIVERSITY OF
BIRMINGHAM

Centre for Liver and Gastroenterology Research
Institute for Immunology and Immunotherapy
Institute for Biomedical Research
The Medical School
University of Birmingham
Birmingham B15 2TT
UK

18th July 2019

Re : Response to referees' critique NCOMMS-18-16530A

Dear Sirs,

I am writing in response to the reviewers critique of our manuscript 'Blocking platelet activation enhances liver recovery after acetaminophen-toxic liver injury by enhancing hepatic neutrophil recruitment : NCOMMS-18-16530A" We are grateful to the editors and referees for the time they have taken to review our original and revised manuscripts and for their continuing insightful critique. It was a pleasure to see that the editorial team considered the study to be of interest for publication and that the majority of the referees now deem the manuscript worthy of publication in Nature Communications.

We note that the editorial team had concerns regarding the mechanism of how neutrophils were protective in the context of acute liver injury and the exact source of our observed TNF- α upregulation. The latest published observations^{1,2} confirm a reparative role for neutrophils in acute liver injury. However, these and other studies stop short of offering viable therapies in acute liver failure. Our work provides the first description of specifically enhancing neutrophil driven liver healing through manipulation of platelet function. Given the multitude of antiplatelet therapy available and already in use, the translatable potential of our work to human acute liver failure is clear.

We have now conducted substantial further investigations and extensively re-written our manuscript; specifically, we now clearly show the TNF- α response on platelet blockade to be macrophage driven and further demonstrate greater mechanistic insights with regards to how neutrophils in CLEC-2 deficiency may be functioning to reduce liver injury. In doing so we believe we addressed the comments from the final referee and detail all of our responses to the remaining queries below.

Reviewer #2 (Remarks to the Author):

The authors have substantially addressed my original concerns. They should consider the following in their revision:

Fig 1a: How long after the human acute APAP poisoning occurred were these livers taken for analysis? Was this a liver biopsy or a section taken after death?

Response: The sections shown were not prepared from a biopsy but were rather pathological specimens collected from explanted livers of patients who were transplanted after they had met the United Kingdom criteria for super-urgent liver transplantation. There exists considerable variability in the time point of APAP overdose and timepoint of transplantation (due to organ availability, time when patient presents to hospital, physiological parameters of recipient etc) and due to the presence of encephalopathy an accurate history with regards to exact timing of overdose often not available. Samples were however likely to be collected within 48-72 hours of presentation to our transplant unit. This has been made clear in the methods (methods; human tissue section).

It seems that the penultimate comment by reviewer 4 regarding Kupffer cells being the source of TNF in vivo has not been addressed. The results stated in the author response do not address whether Kupffer cells are the TNF source in the APAP mouse model. Interference with Kupffer cell production of TNF would be needed to address this. This is not needed for publication of this manuscript, but the authors should be careful about ascribing cause and effect in this regard.

Response: We have now used a more careful cytometric approach to analyze cells isolated from injured livers. This demonstrated that hepatic macrophages (newly infiltrated and resident cells) specifically upregulate TNF- α (Figure 5b). The results section labelled 'Neutrophil recruitment in CLEC-2 deficient animals is dependent on macrophage TNF- α production' has been extensively re-written (changes highlighted) to reflect this (Page 12). We further demonstrate that interference with systemic TNF- α using etanercept abrogates the protective effect of a CLEC-2 deficiency (Figure 5e) by reducing reparative neutrophil recruitment to the injured liver (Figure 5f). We however agree with the referee that hepatic macrophages may not be exclusive source of TNF- α production in acute liver injuries and have thus changed the discussion to reflect this (Discussion; paragraph 2)

Line 140: "human and murine livers respond to acute liver injuries by increasing hepatic podoplanin expression" This statement is too broad and should be confined to what the authors actually showed, i.e., that podoplanin seems to be expressed in APAP overdose injury. Whether it occurs in other injuries in humans is not addressed.

Response: We agree with the referee and have changed text to reflect this.

Line 356: Is there a misplaced comma here (should be after "in vivo")?

Response: We have changed this.

Reviewer #3 (Remarks to the Author):

All issues raised by this reviewer have been sufficiently addressed.

Reviewer #4 (Remarks to the Author):

The authors made effort to revise the manuscript based on reviewers' comments; however, there were only partial revisions. Primary concerns remain. Specifically, 1) the staining of podoplanin in most images looks like in infiltrated macrophages but not Kupffer cells,

Response: It is indeed possible that the cells highlighted represent infiltrated macrophages not resident Kupffer cells. To confirm this we repeated our the toxic injury models and specifically used markers to distinguish between infiltrating macrophages (CD11b^{hi}F480^{int}) and Kupffer cells (CD11b^{int}F480^{hi}) (Figure 1c,d and Supplementary 1)³ and found that whilst both cell types upregulate podoplanin after an acute toxic liver injury, there are greater numbers of infiltrating macrophages present (Figure 1d). Results section 'Podoplanin is upregulated within the liver during liver injury and platelets sequester to podoplanin expressing macrophages' has been re-written to reflect this (changes highlighted).

2) LPS was still used in some in vitro experiments,

Response: LPS was merely used as an artificial *in vitro* stimulus to upregulate podoplanin expression on macrophages, this has been shown previously⁴. The rationale for this was reliably to upregulate podoplanin expression to isolate and study the platelet CLEC-2; macrophage podoplanin (PDPN) interaction, rather than study LPS as an individual model per se. As APAP or CCl₄ require CYP2E1 to be metabolized to toxic intermediates (present in only hepatocytes not macrophages) we could not use these agents to induce macrophage PDPN expression *in vitro*.

and 3) P-selectin is not a reliable maker for platelet activation in tissue sections, as it is constitutively stored within platelets.

Response: P-selectin is indeed stored in platelets, but we would point out that regardless we saw significantly more P-selectin expression in WT animals than CLEC-2 –deficient mice (Figure 2e), still supporting the case for enhanced platelet-dependent effects in this context. We have considered other markers such as TLT-1⁵ as tools to show enhanced local platelet activation within the liver. However, all have the same problems in that they are stored in platelet granules and thus could be detected even in resting platelets. It is noteworthy however that numerous authors have demonstrated that removing or blocking CLEC-2 switches off platelet activation⁶⁻⁸, and the PF4creCLEC1b^{fl/fl} mouse is an established model to study this^{4,9}.

The revised manuscript also still lacks mechanistic insights. No evidence shows that deficiency of platelet CLEC-2 leads to increased TNF in Kupffer cells.

Response: We now show that in APAP induced liver injury, macrophages (both resident and infiltrating) upregulate TNF- α (Figure 5b), and that there are an increased number of these macrophages in CLEC-2 deficient mice (Figure 5c). Taken together this suggests that there would be increased TNF- α within CLEC-2 deficient mice after acute liver injury, we confirm this in Figure 5d where we show that there is a larger amount of TNF- α present within the sera of CLEC-2 deficient mice after acute liver injury. Results section labelled 'Neutrophil recruitment in CLEC-2 deficient animals is dependent on macrophage TNF- α production' has been extensively re-written (changes highlighted) to reflect this.

How infiltrated neutrophils play protective roles in the recovery of APAP-induced liver injury is not studied. This manuscript provides no evidence that TNF is specifically released from the Kupffer cell.

Response: We would disagree somewhat with this statement, as there is now a reasonable body of emergent literature supporting the restorative roles of neutrophils in acute liver injury^{1,2,10}. Defective neutrophil phagocytosis results in poor outcomes in not only acute liver failure¹⁰ but liver disease in general¹¹, we have cited this throughout the manuscript. We now show, cytometric data illustrating that macrophages including both Kupffer cells and infiltrating macrophages upregulate and produce TNF- α in our models (revised Figure 5). However, we agree that all of the TNF- α we see may not be produced by macrophages *in vivo* and thus we have tempered the text to reflect this (Discussion; paragraph 2). Finally, we have undertaken further studies to assess neutrophil function (methods; highlighted section 'Neutrophil activation') and found that neutrophils from mice with a CLEC-2 deficiency exhibited greater phagocytic ability (results section 'Blocking the CLEC-2-podoplanin interaction enhances neutrophil recruitment to the injured liver" page 13 lines 14-24, Figure 5e).

Enhanced neutrophil phagocytosis (Figure 5e) coupled with greater hepatic infiltration (Figure 5b) in CLEC-2 deficient mice, would drive enhanced clearance of apoptotic and necrotic hepatocyte debris in acute liver injury thus accounting for our observations of enhanced liver recovery.

Please do not hesitate to contact me if you require additional information

Yours sincerely

Dr Patricia Lalor
p.f.lalor@bham.ac.uk

- 1 Yang, W. *et al.* Neutrophils promote the development of reparative macrophages mediated by ROS to orchestrate liver repair. *Nat Commun* **10**, 1076, doi:10.1038/s41467-019-09046-8 (2019).
- 2 Jimenez Calvente, C. *et al.* Neutrophils contribute to spontaneous resolution of liver inflammation and fibrosis via microRNA-223. *J Clin Invest* **130**, doi:10.1172/JCI122258 (2019).
- 3 Holt, M. P., Cheng, L. & Ju, C. Identification and characterization of infiltrating macrophages in acetaminophen-induced liver injury. *J Leukoc Biol* **84**, 1410-1421, doi:10.1189/jlb.0308173 (2008).
- 4 Rayes, J. *et al.* The podoplanin-CLEC-2 axis inhibits inflammation in sepsis. *Nat Commun* **8**, 2239, doi:10.1038/s41467-017-02402-6 (2017).
- 5 Smith, C. W. *et al.* TREM-like transcript 1: a more sensitive marker of platelet activation than P-selectin in humans and mice. *Blood Adv* **2**, 2072-2078, doi:10.1182/bloodadvances.2018017756 (2018).
- 6 May, F. *et al.* CLEC-2 is an essential platelet-activating receptor in hemostasis and thrombosis. *Blood* **114**, 3464-3472, doi:10.1182/blood-2009-05-222273 (2009).
- 7 Suzuki-Inoue, K. *et al.* A novel Syk-dependent mechanism of platelet activation by the C-type lectin receptor CLEC-2. *Blood* **107**, 542-549, doi:10.1182/blood-2005-05-1994 (2006).
- 8 Suzuki-Inoue, K. *et al.* Involvement of the snake toxin receptor CLEC-2, in podoplanin-mediated platelet activation, by cancer cells. *J Biol Chem* **282**, 25993-26001, doi:10.1074/jbc.M702327200 (2007).
- 9 Hitchcock, J. R. *et al.* Inflammation drives thrombosis after Salmonella infection via CLEC-2 on platelets. *J Clin Invest* **125**, 4429-4446, doi:10.1172/JCI79070 (2015).
- 10 Taylor, N. J. *et al.* Circulating neutrophil dysfunction in acute liver failure. *Hepatology* **57**, 1142-1152, doi:10.1002/hep.26102 (2013).
- 11 Irvine, K. M., Ratnasekera, I., Powell, E. E. & Hume, D. A. Causes and Consequences of Innate Immune Dysfunction in Cirrhosis. *Front Immunol* **10**, 293, doi:10.3389/fimmu.2019.00293 (2019).

Reviewers' Comments:

Reviewer #2:

Remarks to the Author:

The second revision of the work has addressed my comments adequately.

Reviewer #4:

Remarks to the Author:

The authors made an additional effort to revise the manuscript. However, they stopped short on addressing some important previously asked questions. First, the magnifications/resolution of some key images, such as Figs. 1a/b, 2c, and 3b, are still not sufficient to be interpreted. As was previously pointed out, most podoplanin-positive structures in Fig. 1a/b are likely lymphatic vessels, as the staining pattern does not match those of F4/80- or CD68-positive macrophages in other images. Secondly, P-selectin is not a reliable marker for platelet activation in tissue sections. Therefore, Fig. 2d simply shows there are fewer platelets in the KO liver section. It does not indicate that the KO tissue has fewer activated platelets based on the staining, even though podoplanin/CLEC-2 interaction is known to activate platelets in other settings. Finally, the question of how platelet CLEC-2 suppresses macrophage infiltration and/or their TNF production is not addressed.

The revised manuscript and figures have numerous formatting issues and an inconsistent use of terminology. For example, some images are boxed, some are not; different thicknesses of lines are used; font styles and sizes are inconsistent; CLEC-2 deficient vs CLEC-2 def, Wild Type, WT, and Wild type, and mouse gene names should not be all capitalized, etc.

Dear Professor Conti,

I am writing in response to the reviewers' critique of our manuscript 'Blocking platelet activation enhances liver recovery after acetaminophen-toxic liver injury by enhancing hepatic neutrophil recruitment : NCOMMS-18-16530B". We are grateful to the editors and referees for the time they have taken to review our original and revised manuscripts and for their continuing insightful critique. We are pleased to see that the editorial team remain interested in publishing our study in Nature Communications.

We note there are specific questions from the editorial team and referees. We address these individually below in two sections below.

1) Answer to the Reviewers/referees

Reviewer #2 (Remarks to the Author):

The second revision of the work has addressed my comments adequately.

We thank the reviewer for the time they have taken to review our manuscript.

Reviewer #4 (Remarks to the Author):

a. The authors made an additional effort to revise the manuscript. However, they stopped short on addressing some important previously asked questions. First, the magnifications/resolution of some key images, such as Figs. 1a/b, 2c, and 3b, are still not sufficient to be interpreted. As was previously pointed out, most podoplanin-positive structures in Fig. 1a/b are likely lymphatic vessels, as the staining pattern does not match those of F4/80- or CD68-positive macrophages in other images.

The referee here mentions that the structures in the images 1 a/b represent lymphatic vessels. Across all versions of our submitted manuscript we consistently state this fact in the text (page 6). Specifically, in light of previous queries in past revisions, we have provided both low resolution images to show lymphatic vessel staining in Figure 1 a/b and then high resolution, high magnification images of injured livers to support our conclusions i.e. podoplanin expression on hepatic macrophages in portal and central areas thus image panel(s) 1c, ic-ii, 1f and 2d illustrate this fact. We reiterate that in the uninjured liver we only see expression of podoplanin on lymphatic vessels in the portal area in Zone 1 whilst in the context of disease this expression is increased. This enhanced expression includes F4-80/CD68 positive macrophages in the central areas around the areas of necrosis (zone 2 and 3) and also portal areas. This is further supported by our cytometric data in Figure 1d showing resident and infiltrating macrophages that express podoplanin and Figure 5 which shows TNF- α expression by these podoplanin+ macrophages. For Figure 3b we have now provided higher magnification of the images and further use black arrows to highlight the areas of liver necrosis.

b. Secondly, P-selectin is not a reliable maker for platelet activation in tissue sections. Therefore, Fig. 2d simply shows there are fewer platelets in the KO liver section. It does not indicate that the KO tissue has fewer activated platelets based on the staining, even though podoplanin/CLEC-2 interaction is known to activate platelets in other settings.

This comment relates to whether the platelets are active in the tissue. In addition to figure 2d our data also shows enhanced P-selectin expression in the WT mice after liver injury compared to CLEC-2 deficient mice (Figure 2e and Supplementary Figure 3). We however agree with the referee that (as also discussed in our previous correspondence) using P-selectin as a marker of intrahepatic platelet activation after liver injury has limitations, these are however shared by all other markers of platelet activation including gauging syk phosphorylation or perhaps looking at TLT-1¹ in the injured liver. We have thus added a section to our discussion to illustrate the limitations of various methods to definitively show platelet activation (or lack thereof) within the injured liver (Discussion end of paragraph 2). Importantly as the referee mentions above, the CLEC-2/Podoplanin axis reliably activates platelets in multiple settings. Pertinently, one of the senior authors on this paper (SPW) has clearly demonstrated *in vitro* that podoplanin expressing inflammatory macrophages activate platelets via CLEC-2 and that CLEC-2 deficient platelets do not activated in this situation².

c. Finally, the question of how platelet CLEC-2 suppresses macrophage infiltration and/or their TNF production is not addressed.

As per the suggestion of the editorial board we have included this in the discussion as an exciting area that merits further research.

d. The revised manuscript and figures have numerous formatting issues and an inconsistent use of terminology. For example, some images are boxed, some are not; different thicknesses of lines are used; font styles and sizes are inconsistent; CLEC-2 deficient vs CLEC-2 def, Wild Type, WT, and Wild type, and mouse gene names should not be all capitalized, etc.

These formatting inconsistencies have been corrected in the latest version of our submitted manuscript. We apologise for these.

2) Editorial board suggestions

a) Please note that in separate email correspondence, reviewer #4 suggested that staining with an antibody against phospho-Syk (Y525/526) (CST - Clone C87C1) in 4% PFA fixed tissue cryosections could be used as an alternative to P-selectin staining to assess platelet activation. We would recommend that you follow this reviewer's suggestion in order to address his/her repeated concern regarding your P-selectin stainings. confirm differences in platelet activation through methods other than P-selectin staining, specifically referee #4 suggests that staining with an antibody against phospho-Syk (Y525/526) (CST - Clone C87C1) in 4% PFA fixed tissue cryosections could be used as an alternative to P-selectin staining to assess platelet activation.

This comment relates to whether the platelets are active in the tissue. To address this, the referee has suggested the use of a phospho-Syk antibody. However, while we are

grateful for this suggestion, there are several reasons why this would not be suitable including the presence of Syk in other haematopoietic lineages (notably macrophages) and because of 'off-target' binding to other sites of phosphorylation. While it is possible to use the antibody to monitor Syk phosphorylation on 525/526 by gel chromatography, this cannot be used as a stain as the antibody detects other bands on the gel, which are can be overlooked in the gel because of the molecular weight difference. This does not apply however to immunochemistry. This issue also applies to other phosphospecific antibodies especially as the nature of the epitope in the gel and when studied by immunochemistry may not be the same. The only true control in such studies is to knockout the tyrosine residue but then the platelets will not be activated. As mentioned in our previous reply, we cannot think of a way to show that the platelets are activated other than the combination of CD41 and P-selectin staining. However, this conclusion is also supported by the functional data in the manuscript. Specifically, we show a clear difference between CLEC-2 deficient and wild type mice after acute liver injury (Figure 2e and Supplementary Figure 3). Finally previous *in vitro* work from one of the senior authors on this paper (SPW) has shown that inflammatory macrophages upregulate podoplanin and then activate platelets via CLEC-2²; we make further specific references to this work and specify the limitations of attempting to illustrate intrahepatic platelet activation *in vivo* or *ex vivo* after acute liver injury in our revised manuscript (Discussion, end of paragraph 2)

- b) ***correct any inconsistencies in formatting and terminology, as highlighted by this reviewer. Please note we will not be able to consider a revised manuscript until all these points are addressed. We would however not request the additional concern of reviewer #4 regarding the lack of mechanistic insights into how platelet CLEC-2 suppresses macrophage infiltration and TNF production to be experimentally addressed. Instead, a caveat should be added in the main text that these issues would warrant further investigation.***

We have carefully read and corrected the manuscript for inconsistencies in formatting and terminology. We apologise for the inconsistencies. We have added text in the discussion (page 15) to highlight that further studies to address the mechanism of heightened TNF production in CLEC-2 deficient animals are required. We thank the editorial board for this suggestion.

- c) ***For your information we also consulted separately with reviewer #3, who indicated that platelet-leukocyte aggregates in the blood, assessed by flow cytometry, could also potentially reflect differences in platelet activation in the liver, although this reviewer pointed out that the absence of aggregates in the blood may not allow to fully rule out local platelet activation in the liver.***

Measuring circulating platelet-leukocyte aggregates would at best be surrogate measures of intra-hepatic platelet activation and as reviewer #3 notes, lack of or presence of these will not definitively measure local hepatic platelet activation (or lack thereof). Another possibility we assessed was the possibility of flow cytometric staining using CD41 coupled with F4/80 or CD11b from the liver digests of mice that had undergone an acute liver

injury; however this would again not show platelet activation, merely binding or perhaps phagocytosis of platelets by cells of the myeloid lineage-again highlighting the difficulty in definitively showing intrahepatic platelet activation within the acutely injured liver. We have added a section in the discussion describing platelet activation in the injured liver in which we also highlight some of the issues of definitively demonstrating intrahepatic platelet activation including limitations of using P-selectin and other markers (Discussion paragraph 2).

Please do not hesitate to contact me if you require additional information.

Thanking you,
Yours Sincerely,
Dr Abhishek Chauhan
Professor Steve Watson
Dr Trish Lalor

- 1 Smith, C. W. *et al.* TREM-like transcript 1: a more sensitive marker of platelet activation than P-selectin in humans and mice. *Blood Adv* **2**, 2072-2078, doi:10.1182/bloodadvances.2018017756 (2018).
- 2 Kerrigan, A. M. *et al.* Podoplanin-expressing inflammatory macrophages activate murine platelets via CLEC-2. *J Thromb Haemost* **10**, 484-486, doi:10.1111/j.1538-7836.2011.04614.x (2012).

Reviewers' Comments:

Reviewer #4:

Remarks to the Author:

No new comments.

Point by point rebuttal

REVIEWERS' COMMENTS:

Reviewer #4 (Remarks to the Author):

No new comments.

We thank the reviewer for his/her time.

Comments by reviewer 3

- I agree that lymphatic vessels had been indicated clearly in previous versions of the paper, thus, no changes are required in this regard.

- Macrophage expression of podoplanin has been resolved.

- I agree that it is difficult to provide unequivocal evidence for platelet activation in tissues. This includes P-selectin but also Syk phosphorylation and other markers. However, it is somewhat disappointing that the authors have not tried to further address this point experimentally. Combining various "surrogate markers" of platelet activation would support the conclusions. But yes, showing hepatic platelet activation comes with the limitations highlighted by the authors.

- Figure 2d/e is indeed problematic. I very much agree with the reviewer#4 that Figure 2d "simply shows there are fewer platelets in the KO liver section". I must say that I have previously overlooked this problem, because P-selectin (platelet activation marker) and CD41 (platelet marker) are not shown next to each other. In fact, all platelets seem to express P-selectin in this figure. Thus, the reduction in P-selectin on respective tissue sections is driven by reduced numbers of platelets. Figure 2e is merely a quantification of Figure 2d. Axis labelling "P-selectin expression in %" is misleading because it does not indicate the comparative value it relates to (% of what?). In the literature, P-selectin expression typically relates to platelets. However, in their quantification authors refer to the section area. Thus, the reduced number of platelets on quantified sections will cause less % P-selectin. Authors would need to compare sections with similar platelet coverage and then compare P-selectin expression. As it stands, authors do not provide conclusive evidence for platelet activation in liver tissues.

To address this we have removed all claims of platelet activation in the figure and changed to reduced platelet number both in this figure, related text and supplementary figures.

- I very much agree with the editor that the paper would add interesting findings to the field of liver immunology and hepatic regeneration. Genetic targeting of CLEC-2 and especially the blocking antibody offer a new therapeutic strategy in the inflammatory conditions studied.

- There is strong and ample evidence of the role of CLEC-2 as a platelet-activating receptor. Thus, to me the degree of platelet activation is not the key message here. However, textual changes would need to be made to tone down the term "platelet activation"

We thank the reviewer for his/her time and have made all the textual changes as suggested.